# Dissecting the properties of circulating IgG against streptococcal pathogens through a combined systems antigenomics-serology workflow

Alejandro Gomez Toledo [1,5], Sounak Chowdhury[1,5], Elisabeth Hjortswang[1], James T. Sorrentino[2], Nathan E. Lewis [3], Anna Bläckberg[1], Simon Ekström [4], Sven Kjellström [4], Arman Izadi [1], Berit Olofsson [1], Pontus Nordenfelt [1], Lars Malmström [1], Magnus Rasmussen[1] & Johan Malmström [1,4] ✉

This study showcases an integrative mass spectrometry-based strategy combining systems antigenomics and systems serology to characterize human antibodies in clinical samples. This strategy involves using antibodies circulating in plasma to affinity-enrich antigenic proteins in biochemically fractionated pools of bacterial proteins, followed by their identification and quantification using mass spectrometry. A selected subset of the identified antigens is then expressed recombinantly to isolate antigen-specific IgG, followed by characterization of the structural and functional properties of these antibodies. We focused on Group A streptococcus (GAS), a major human pathogen lacking an approved vaccine. The data shows that both healthy and GAS-infected individuals have circulating IgG against conserved streptococcal proteins, including toxins and virulence factors. The antigenic breadth of these antibodies remains relatively constant across healthy individuals but changes considerably in GAS bacteremia. Moreover, antigen-specific IgG analysis reveals individual variation in titers, subclass distributions, and Fc-signaling capacity, despite similar epitope and Fc-glycosylation patterns. Finally, we show that GAS antibodies may cross-react with *Streptococcus dysgalactiae* (SD), a bacterial pathogen that occupies similar niches and causes comparable infections. Collectively, our results highlight the complexity of GAS-specific antibody responses and the versatility of our methodology to characterize immune responses to bacterial pathogens.

Immunoglobulin G (IgG) is a central effector molecule of adaptive immunity that leverages protective responses against microbial infections. IgG binds to the surface of viral and bacterial pathogens and to soluble toxins, to neutralize their capacity to damage host tissues. Neutralization is mediated by the fragment antigen-binding (Fab) region, which recognizes epitopes on microbial proteins and polysaccharides. Neutralizing Fab binding prevents key steps in the establishment of an infection, including pathogen adhesion and cellular invasion. Besides neutralization, antigen-bound IgG can also trigger the initiation of the classical complement pathway, as well as

---

other protective responses, such as antibody-dependent cellular cytotoxicity (ADCC) and antibody-dependent cellular phagocytosis (ADCP)[1]. These effector functions are finetuned by the structure of the fragment crystallizable (Fc) region, especially by the Fc subclass and glycosylation, which synergistically modulate the IgG affinity for complement and immune cell receptors[2].

During antimicrobial responses, polyclonal IgG targets several antigens on a given pathogen, and various epitopes within each antigen, resulting in a broader range of protective responses compared to monoclonal IgG. However, characterizing the properties of polyclonal antibodies at a systems-wide level, including their antigenic repertoires, binding epitopes, subclass distributions, glycosylation patterns, and effector functions, remains a significant analytical challenge[3]. In turn, a poor understanding of the structural and functional IgG features that contribute to host protection prevents the identification of useful correlates of immunity to major human pathogens and the development of antimicrobial vaccines[4,5].

Recently, efforts in reverse vaccinology have led to the development of systems antigenomic approaches that exploit the availability of annotated genome data, novel surface display technologies, and proteomics workflows, to characterize microbial antigens recognized by antibodies and T-cells. Examples include screening genome sequences of Group B Streptococcus, cloning surface-exposed antigens, and conducting immunization challenges in animal models; building protein arrays paired with flow cytometry binding assays to study antibodies against predicted surface GAS proteins; and using reverse vaccinology and human infection challenge models to explore the antigenic breadth of antibodies against the malarial parasite *Plasmodium falciparum*[6–12]. Systems antigenomics has been successful in defining pathogen-specific antibody antigenomes, (i.e., the spectrum of molecules expressed by a given pathogen that are recognized by host antibodies), a central bottleneck of most vaccine development pipelines[13,14]. However, with an obvious focus on antigen identification, systems antigenomics does not inform on other antibody properties beyond Fab binding.

Advances in Omics technologies have also sparked the field of systems serology, a collection of integrative approaches to analyze various antibody features and functions, coupled with advanced computational and statistical methods[12,15–17]. Systems serology has been useful to deconvolute immune correlates of protection and vaccine efficacy for the Human Immunodeficiency Virus (HIV)[18], *Mycobacterium tuberculosis* (MTB)[19], and SARS-CoV-2[20,21]. These studies have revealed that humoral responses elicited in four HIV vaccine trials result in unique humoral "Fc fingerprints", that individuals with latent tuberculosis infection and active tuberculosis disease exhibit distinct MTB-specific humoral responses, and that specific Fcγ-receptor signaling plays a role in controlling SARS-CoV-2 infections, to only mention a few examples. On the other hand, the starting point of systems serology is typically one or a few preselected antigen(s), a choice that often relies on previously acquired knowledge.

Mass spectrometry (MS) is a highly sensitive and versatile analytical method for protein identification, quantification, and the characterization of post-translational modifications and protein-protein interactions. Despite advances in systems antigenomics for unbiased antigen discovery and systems serology for functional antibody analysis, these methods are often applied independently, leaving critical gaps in linking antigen specificity with functional antibody responses. The flexibility of modern MS technologies now allows for the integration of these complementary approaches. Here, we present an integrated, automated, and quantitative workflow that combines systems antigenomics and systems serology to provide an extensive, multilayered characterization of antigen-specific antibodies. Compared to previous methods, this workflow identifies relevant antigens in an unbiased manner while simultaneously analyzing antibody titers, epitope landscapes, subclass distributions, and Fc-mediated functions.

We applied this approach to study antibody responses against Group A Streptococcus (GAS) (*Streptococcus pyogenes*; *S. pyogenes*), a major bacterial pathogen and significant source of global morbidity and mortality worldwide[22].

## Results

Most adult individuals have circulating IgG antibodies against GAS, but their structural and functional properties remain poorly understood[23]. To address this challenge, we developed a two-step approach to (i) determine the GAS-antigenome, and (ii) characterize the titers, epitope repertoires, immune signaling capacity, subclass distributions, and N-linked glycosylation profiles of the antigen-specific IgG. The approach is based on streamlining antigen/antibody affinity purification workflows using an automated liquid-handling platform, coupled to a suite of high-resolution MS-based quantitative, structural, and glyco-proteomics readouts. The first step (systems antigenomics) focuses on antigen discovery, leveraging antigen/antibody affinity purification to isolate and identify GAS-specific antigens from bacterial protein fractions using high-resolution mass spectrometry. In the second step (systems serology), we focus on a subset of these identified antigens for detailed characterization of the corresponding antibody responses, integrating systems serology techniques to analyze functional and structural properties of the antigen-specific antibodies. The antigenome data then informs the selection of key antigens, enabling a focused analysis of their associated antibody responses. An overview of the main components of this analytical strategy is shown in Fig. 1a.

### Mapping the GAS-antigenome

To define the GAS-antigenome, we exploited GAS-specific IgG circulating in human plasma as a tool to isolate antigens from pools of bacterial proteins via affinity purification coupled to LC-MS/MS (Fig. 1b). First, the SF370 strain, of the clinically relevant M1 GAS serotype, was biochemically fractionated into defined pools of potentially antigenic proteins (Supplementary Fig. 1). The SF370 strain was the first fully sequenced strain of *S. pyogenes* and has served as a reference for studies exploring both GAS antigens as well as mechanism of GAS disease[24]. The identity and cellular localization of bacterial proteins in pools from a typical preparation are presented in Fig. 1c, d and Supplementary data 1. We primarily focused on surface-exposed and secreted proteins since they are more likely to be recognized by host antibodies. Next, we used two IgG sources to isolate antigens from these bacterial fractions: (i) pharmaceutical-grade intravenous immunoglobulin G (IVIG), and (ii) commercial pooled human plasma (HP). Both IVIG and HP contain IgG antibodies from many different individuals, including clones directed against multiple GAS antigens (Fig. 1e) and with the capacity to induce protective opsonophagocytic responses (Fig. 1f). IgG was then immobilized on Protein G columns to enrich for GAS antigens from the subcellular fractions. Retained antigens were eluted and identified by LC-MS/MS. Positive identifications were required to have at least a 2-fold enrichment over background levels, where reactive polyclonal antibodies were substituted with control IgG of irrelevant specificity (Xolair, anti-human IgE).

This antigenomics approach identified a total of 39 antigens: 13 were unique to the IVIG, 2 were unique to HP, and 24 were common to both IVIG and HP (Fig. 1g, h and Supplementary data 2). The method was highly reproducible and generated a similar number of identifications across replicates, with a broad distribution across the dynamic range of the input proteome of the bacterial fractions and no obvious bias against high or low-abundant proteins (Supplementary Fig. 2). The GAS antigenome was enriched in virulence factors, including toxins (e.g., SLO), superantigens (e.g., SPEC, MEZ), anti-phagocytic proteins (e.g., M1), and enzymes (e.g., C5AP, HYLA). A few proteins of unknown function (e.g., PRGA) or without an obvious link to GAS virulence (e.g., ribosomal proteins RPLA and RPSB) were also identified (see Dataset

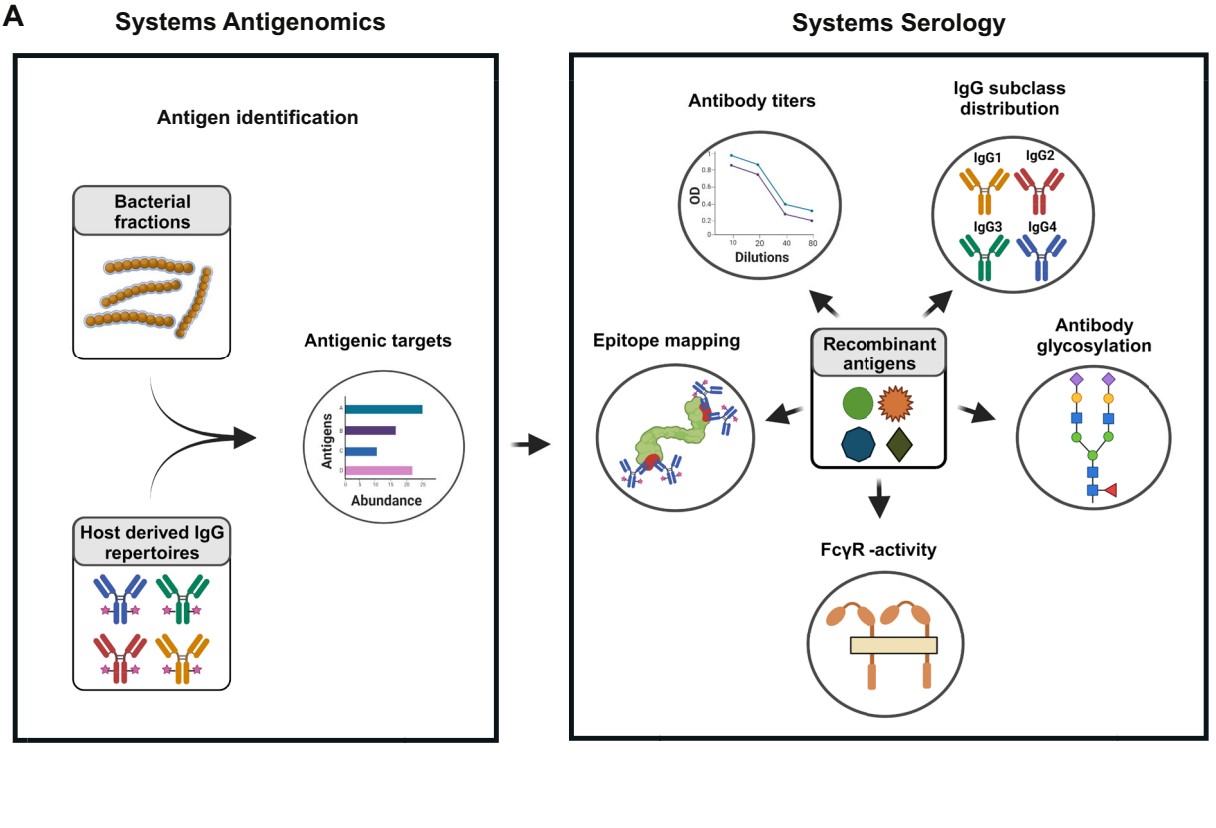

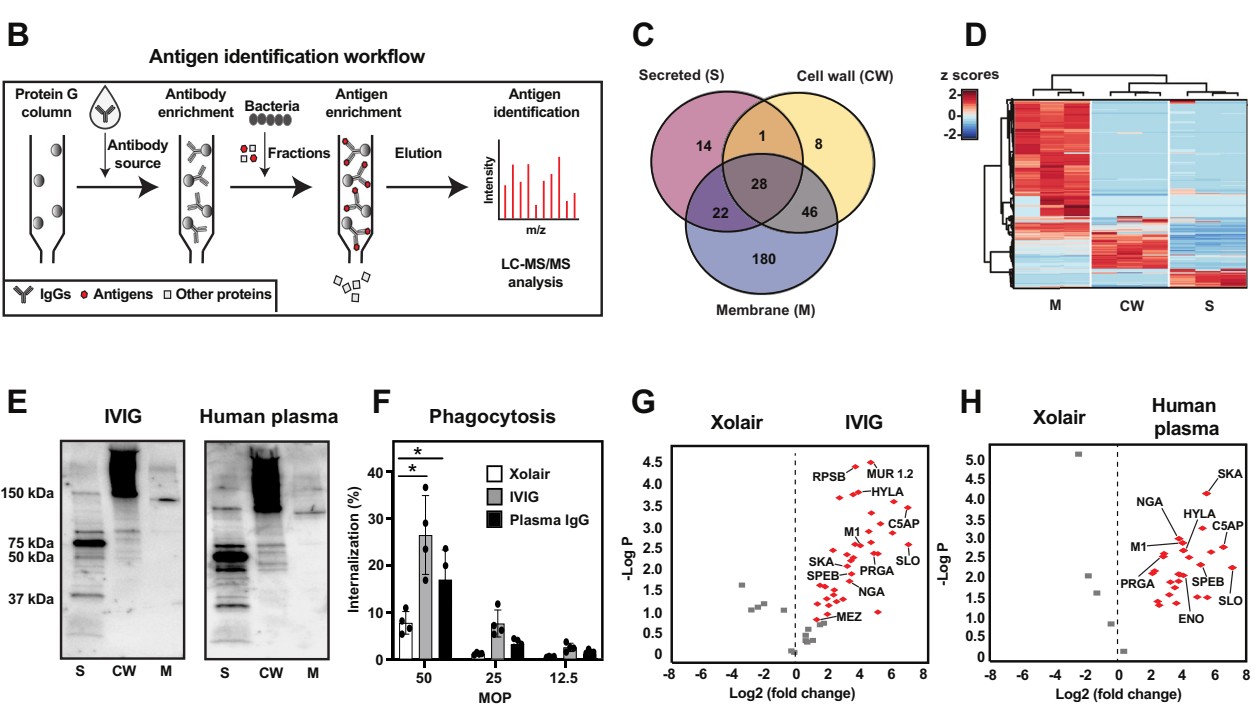

S2, which includes descriptions of the acronyms for the selected GAS proteins used throughout the manuscript).To evaluate whether these results were serotype-specific, two additional GAS strains were cultured, fractionated, and subjected to the same antigenomics analysis: AP1, a more virulent M1-serotype than the SF370[25], and one M49 strain, expressing a shorter M-protein and associated with a different set of clinical manifestations[26]. In general, serotype-specific proteins such as the M-protein (M1 vs M49) were recovered using both IVIG and human plasma, indicating that the workflow could be extended to characterize antibodies against pathogens with different antigen repertoires

(Supplementary Fig. 3). Despite a significant overlap, unique antigens associated with these new strains were also identified, including Serum Opacity Factor (SOF), a protein known to be expressed by M49 but not M1 GAS serotypes[26], and accordingly identified only in samples using M49-derived proteins. Combined, the results from all examined 3 GAS strains accounted for the identification and quantification of 68 unique antigens recognized by IVIG and pooled human plasma. At least 11 of these antigens have previously been shown to confer protection against GAS, either in vitro or in vivo (Supplementary data 2). We concluded that GAS-specific IgG circulating in human plasma targets a

**Fig. 1 | The GAS-specific IgG antigenome. a** Schematic representation of the two-step approach integrating systems antigenomics and systems serology. The systems antigenomics strategy involves the identification of antigenic targets from biochemical fractions of bacterial proteins using host-derived IgG as a guide. Selected antigens are then recombinantly produced and analyzed in a streamlined workflow of various systems serology techniques to deconvolute structural and functional attributes of the antigen-specific IgG. Created in BioRender. Malmström, J. (2025) https://BioRender.com/s45i129. **b** Schematic summary of the antigen identification workflow used in this study. **c** Overlap of the bacterial proteins identified across secreted (S), cell wall (CW), and membrane (M) fractions during a typical biochemical fractionation of the SF370 GAS proteome. **d** Differential protein expression across bacterial fractions. The protein values were normalized using a Z-score normalization and subjected to hierarchical clustering.

**e** Immunoblot analysis of antigens in S, CW, and M fractions of the SF370 GAS strain using IVIG and pooled human plasma. Immunoblots are representative images of at least 2 independent experiments. **f** Phagocytosis of SF370 bacteria with Xolair, IVIG, and IgG purified from pooled human plasma. Bar graphs represent percentage of THP-1 cells with internalized bacteria for different multiplicity of prey (MOP-ratio of prey to phagocyte)—50, 25, and 12.5. Four technical replicates were used for each condition and statistical significance was assessed by two-way ANOVA with Tukey's multiple comparisons test, $*p < 0.05$. Bars represent mean values and error bars represent standard deviation (SD). **g** Volcano plot displaying the significant antigens recognized by IVIG and (**h**) pooled human plasma. Statistical significance was determined using a both side $t$-test with an FDR of 0.05 to correct for multiple comparisons. Source data are provided as a Source Data file.

small subset of bacterial antigens, many of which are well-known virulence determinants and potential vaccine candidates.

## The GAS-antigenome is conserved across healthy individuals but is altered in GAS bacteremia

In the next step, we analyzed plasma from 10 healthy donors to investigate potential individual variations in the GAS antigenome. In addition, both acute and convalescent plasma from four patients with GAS bacteremia were included to determine whether invasive infections might affect the antigenome profiles. We have previously reported the clinical and serological status of these patients, two of the patients were infected with emm1 isolates, and two with emm118 or emm85, but no major differences were found between their acute and convalescent plasma[27]. The antigenome analysis using fractions from the SF370 GAS strain resulted in the identification of a total of 72 antigens, with an average of ~30 antigens/individual (Supplementary data 3). A substantial overlap was observed across the individual antigenomes. The level of antigen enrichment varied across samples and correlated with antibody titers measured by ELISA, as evaluated for two antigens: C5AP and PRGA (Supplementary Fig. 4). The antigenome profiles were rank correlated and linked by network analysis using Kendall Tau coefficients (Fig. 2a). The networking approach segregated the data into two distinct clusters. Cluster A was driven by antigens enriched in the patients with GAS bacteremia (e.g., ENO, LDH) (Fig. 2b), whereas Cluster B was driven by antigens enriched in the healthy group (e.g., MF, PRSA1) (Fig. 2c). However, independently of disease status, all individuals had circulating antibodies against a common set of 11 antigens, suggesting a potential immunological signature of GAS exposure (Fig. 2d). These antigens were primarily GAS virulence factors involved in pathogenesis and immune evasion (e.g., SLO, M1, C5AP, SPEB etc.).

Finally, the amino acid sequence for each of the 72 antigens was compared across 2275 publicly available GAS genomes, which revealed high gene carriage, based on their presence in >90% of all genomes, and high sequence conservation, based on low sequence entropy and gap occurrences. One notable exception was the M1 protein, due to the high sequence variability of the hypervariable region (HVR) (Fig. 2e)[28]. In summary, and similar to pooled samples, the individual antigenomes converged around a small subset of genomically conserved antigens. These profiles were similar across healthy individuals but were considerably different in patients with GAS bacteremia.

## The GAS antigenome has sequence homologues in the antigenome of *Streptococcus dysgalactiae*

GAS colonizes similar host tissues as *Streptococcus dysgalactiae* (SD), a gram-positive bacterium commonly found in the human throat and skin[29]. While SD colonization is usually asymptomatic, it can sometimes cause various clinical conditions, such as pharyngitis, impetigo, and sepsis. GAS and SD have significant genomic similarities, largely due to horizontal gene transfer (HGT)[30,31]. This genomic relationship results in high homology between some of their virulence factors,

raising the possibility of immunological crosstalk between the GAS and SD antigenomes.

To explore this possibility, we mapped the sequences of the 72 antigens identified as part of the GAS antigenome against 268 unique SD genomes. A substantial portion of these GAS antigens had homologous counterparts in SD with more than 50% sequence identity (Fig. 3a). However, certain GAS antigens, such as SPEB and IdeS, did not have homologous sequences in SD. In contrast, other antigens such as M1, SLO, SKA, and C5AP exhibited high sequence homology to SD genes, but these homologues were only present in about 50% of the analyzed SD genomes. Most GAS antigen homologues were highly conserved across SD genomes and displayed low sequence variation (Fig. 3b), with the notable exception of the M proteins. Similar to the GAS M proteins, the SD M proteins are highly polymorphic in the N-terminus, and sequencing of this hypervariable region constitutes the basis for pathogen serotyping[29].

Next, we subjected a clinically relevant SD isolate of the stG62647 serotype[32] to antigenomics analysis and compared the SD antigenome recognized by IVIG and HP, to the SF370 antigenome recognized by these same pools of antibodies. On average, 28 SD antigens were uniquely identified using IVIG, 22 using HP, and 19 were common to both (Fig. 3c, d and Supplementary data 4). Notably, 11 of the antigens that displayed substantial sequence homology between the species (>50%) were significantly enriched from both GAS and SD lysates using IVIG, and 8 using HP (Fig. 3e–g). Interestingly, some antigens, such as CH60, displayed high homology across species (90–100%) and large percentage fold changes compared to control samples, indicating that CH60 might be immunologically cross-reactive. A similar observation applied to GAS SPY0469 and the SD GBS_IP (50–100%). However, other homologous pairs, such as streptokinase (SKA) (80–90%), were more enriched using GAS lysates compared to SD bacterial fractions, suggesting either a difference in their protein levels in the original bacterial lysates or that the antibodies might differentially target non-homologous epitopes. Well-known GAS virulence factors, such as SLO, C5AP, NGA, and SPEB were uniquely assigned to the GAS but not to the SD antigenomes. These antigens were also exclusively identified in the original GAS but not in SD bacterial fractions (Fig. 3h, i). This result demonstrates that these antigens were either not present or at least not produced by the stG62647 clinical isolate, aligning with bioinformatics analysis in Fig. 3a showing their complete absence or lower carriage across the sequenced SD genomes.

Conversely, other antigens, such as the M proteins, which display a sequence homology of 55% across these streptococcal species, were highly abundant in both GAS and SD bacterial lysates (Fig. 3j and Supplementary Fig. 1). However, in sharp contrast to the SF370 M1 protein, the stG62647 M protein was not significantly enriched when using IVIG or HP (Fig. 3j). Since the M1 and the stG62647 M proteins are sequence divergents towards the N-terminal part, but highly conserved towards the C-terminus (Fig. 3l), this differential enrichment of the M-proteins suggests that the epitopes targeted by the M-specific antibodies in IVIG and HP are directed towards non-homologous and

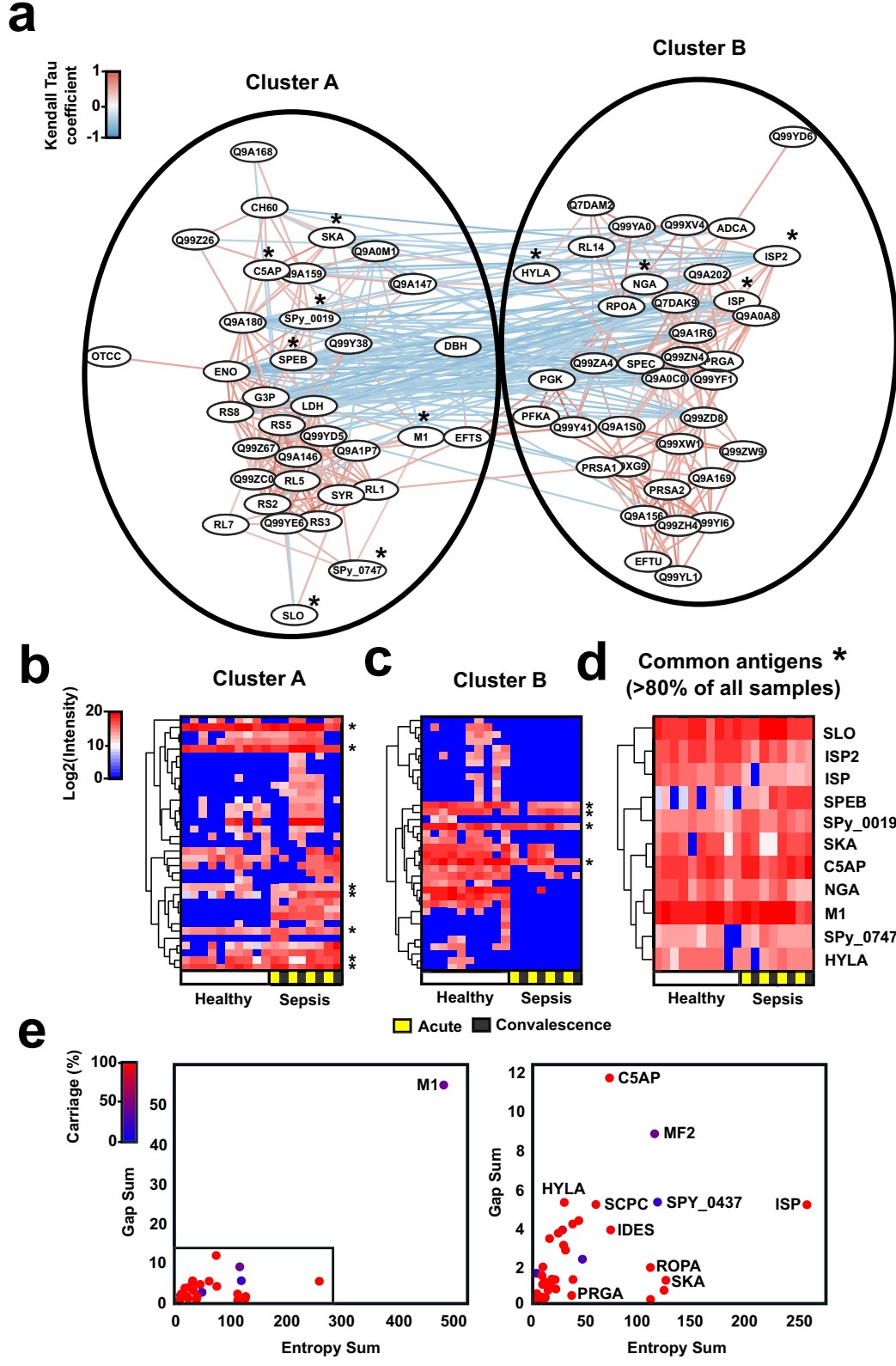

therefore species-specific regions, i.e., type-specific antibodies directed against GAS but not SD. In contrast to the GAS M1, the stG62647 M protein was not enriched when antigenomic screening was conducted using individual plasma from five different healthy individuals or convalescent plasma from five SD septic individuals who were infected by the exact same stG62647 clinical isolate (Fig. 3k). Together,

these observations strongly suggest the lack of abundant circulating antibodies against the stG62647 M protein in the analyzed samples. Finally, the overall SD antigenomes of the septic patients could not discriminate between healthy and infected individuals, indicating that changes in antigenome profiles during sepsis are likely pathogen-specific (Supplementary Fig. 5 and Supplementary data 5).

**Fig. 2 | The GAS-specific IgG antigenome across healthy and sepsis individuals.**
**a** Rank correlation network of the 72 GAS antigens identified across different individuals. Nodes represent each identified GAS antigen and the distance between the nodes is defined by edges encoding Kendall tau coefficients for each pairwise comparison. The color of the edges reflects positive (red) or negative (blue) correlation coefficients. To be considered part of the antigenome, the proteins were required to be present in at least two out of three biological replicates, and identified by at least two quantifiable peptides, having at least a two-fold enrichment over the negative control (Xolair, a commercial anti-IgE monoclonal). Pearson correlation clustering of the log2 intensity of the antigens in (**b**) Cluster A, (**c**) Cluster B, and (**d**) the 11 common antigens across healthy and sepsis individuals. **e** Sequence conservation plots of the 72 antigens based on the analysis of gap frequency, entropy, and gene carriage of each protein across 2275 GAS genomes (left), and zoom in plot of the antigens excluding M1 (right). Residues with high conservation have low entropy, whereas residues with low conservation have high entropy. Gaps indicate insertion and deletion in sequences.

In conclusion, our data highlight both genomic and antigenic similarities between GAS and SD but also reveal significant differences in pathogen-specific antibody responses during sepsis.

## Mapping antigenic sites frequently targeted by circulating GAS-antibodies

The antigenome analysis identified a specific set of antigens commonly targeted by circulating GAS antibodies. However, polyclonal IgG can bind to multiple epitopes within a single antigen, potentially leading to differences in biological outcomes. The differential enrichment of the GAS M1-protein compared to its homologous SD counterpart, further highlighted the importance of characterizing both the targeted antigens and their specific epitope(s), to fully understand the capacity of GAS-antibodies to exert immune functions.

To determine the epitope landscape of GAS-specific IgG, we implemented an epitope extraction (EpXT) workflow (Fig. 4a and "Materials and Methods"). We selected three antigens for this analysis: M1 and C5AP, identified in Cluster A, and PRG4, identified in Cluster B (Fig. 2a). M1 and C5AP were chosen because they are well-known virulence factors and vaccine candidates. In contrast, the function of PRG4 remains unknown, providing an opportunity to explore a less characterized protein. Structurally, M1 is a fibrillar protein, C5AP is a globular protein, and PRG4 is predicted to contain both fibrillar and globular regions, offering a diverse range of antigenic structures for analysis.

All three proteins were recombinantly expressed and subjected to limited proteolysis to generate partially digested protein regions of different sizes. The partial digests were captured by immobilized IVIG antibodies to isolate antigenic protein regions, which were eluted and quantified by LC-MS/MS. The method was first applied to C5AP, a streptococcal serine peptidase with a multidomain structure: a protease-associated domain (PA-domain), a catalytic domain (Cat-domain), and three consecutively arranged fibronectin-type domains (FN-domains) (Fig. 4b)[33]. The EpXT analysis identified 17 immunogenic peptides (Supplementary data 6). Roughly 70% of the total peptide intensity was associated with the Cat-domain, ~15% with the FN1-domain, and only ~5% with the FN2 domain (Fig. 4b). To validate these immunodominant regions, we performed hydrogen-deuterium exchange mass spectrometry (HDX-MS), a technique that measures the exchange of hydrogen atoms with deuterium in a protein's backbone amide groups. The rate of this exchange is influenced by the structural flexibility and solvent accessibility of the protein, with regions buried within the protein structure or bound by antibodies exhibiting slower exchange rates. HDX-MS identified two peptide stretches (97–138 aa and 415–466 aa) that displayed a significant reduction in deuterium uptake upon incubation with IVIG (Fig. 4c and Supplementary data 7 and 8), indicating that these regions were engaged by antibody binding. These binding sites partially overlapped with those identified by the EpXT workflow, demonstrating good agreement between the methods and further validating the Cat-domain as an immunodominant region (Fig. 4d).

The interaction between IVIG and the M1-protein was also studied by EpXT. M1 is a dimeric coiled-coil fibrillar protein with an N-terminal HVR, a variable region encompassing the A domain and B repeats, and a constant region comprising the C repeats and D domain. We identified 23 antigenic peptides across the three regions. Roughly 26% of the intensity was associated with the HVR, 52% with the variable region and only 22% with the constant region (Fig. 4e). These results demonstrate that HVR and the variable region are the most commonly targeted and immunodominant antigenic sites, providing a molecular explanation for the lack of cross-reactivity of M1-specific antibodies with the C-terminal homologous M-protein expressed by SD (Fig. 3j–l). To address whether similar epitopes are recognized by IgG from different individuals, we interrogated plasma from healthy individuals, as well as the paired acute and convalescent plasma from patients with GAS bacteremia. The individual epitope patterns were consistent with the pattern observed when using IVIG, both in terms of the peptide identities and their relative intensity distributions (Fig. 4e). Overall, the epitope profiles were similar across healthy individuals, and between the acute and convalescent plasma of each patient, although peptides from the constant region of M1 tended to be more enriched when using plasma from patients with bacteremia. Notably, we identified five overlapping peptides (aa 257–277, 262–277, 262–284, 299–319, and 304–319) from the constant region that were differentially enriched. Peptides 257–277 and 299–319 were substantially more enriched compared to the other 3 and differed only by an additional five N-terminal amino acid stretch (KQVEK). These results suggest that KQVEK is a critical part of the epitope targeted by these antibodies. Furthermore, averaging the peptide signal across all samples reinforced that the HVR and the variable region are the most commonly targeted and immunodominant sites (Fig. 4f).

Finally, PRGA, an 873-a-long GAS protein of unknown function was also analyzed by EpXT. Since there is no structure available, a molecular model of PRGA was generated, which predicted an extended coiled-coiled structure with an internal globular domain (Supplementary Fig. 6). A total of 6 peptides were identified, with ~90% of the intensity associated with the globular domain, thereby also indicating antibody binding to spatially confined and most likely immunodominant regions of PRGA (Supplementary data 6). In conclusion, we show that EpXT can map immunodominant antigenic sites on various GAS antigens. For the M1-protein, most of the immunodominant antigenic sites are located in the HVR and the variable part. Some of these sites are shared across individuals, suggesting common mechanisms of epitope recognition or exposure to common bacterial serotypes.

## The IgG subclass impacts the ability of anti-M1 antibodies to trigger immune signaling

In addition to neutralization through antigen and epitope recognition, antibodies can elicit protective effector functions that are dependent on other IgG properties, such as the Fc-glycosylation and the IgG subclass distribution[2]. To test whether GAS antibodies trigger Fc-dependent immune signaling, recombinant M1 was incubated with IVIG or plasma, and probed for the activation of FcγR-receptor IIa (CD32), a surrogate for ADCP, and IIIa (CD16), a surrogate for ADCC, using luciferase reporter cell assays. The dose-response analysis demonstrated the assay had high sensitivity, and the activity tightly correlated with the titers of M1-specific antibodies in each sample (Supplementary Fig. 7). Next, we tested the ability of IVIG to trigger immune signaling using recombinant C5AP and PRGA. In general, antibodies against all antigens elicited both CD32 and CD16 activation, with significant variation observed across the antigens (C5AP > PRGA > M1) (Fig. 5a, b). These differences could not be explained by

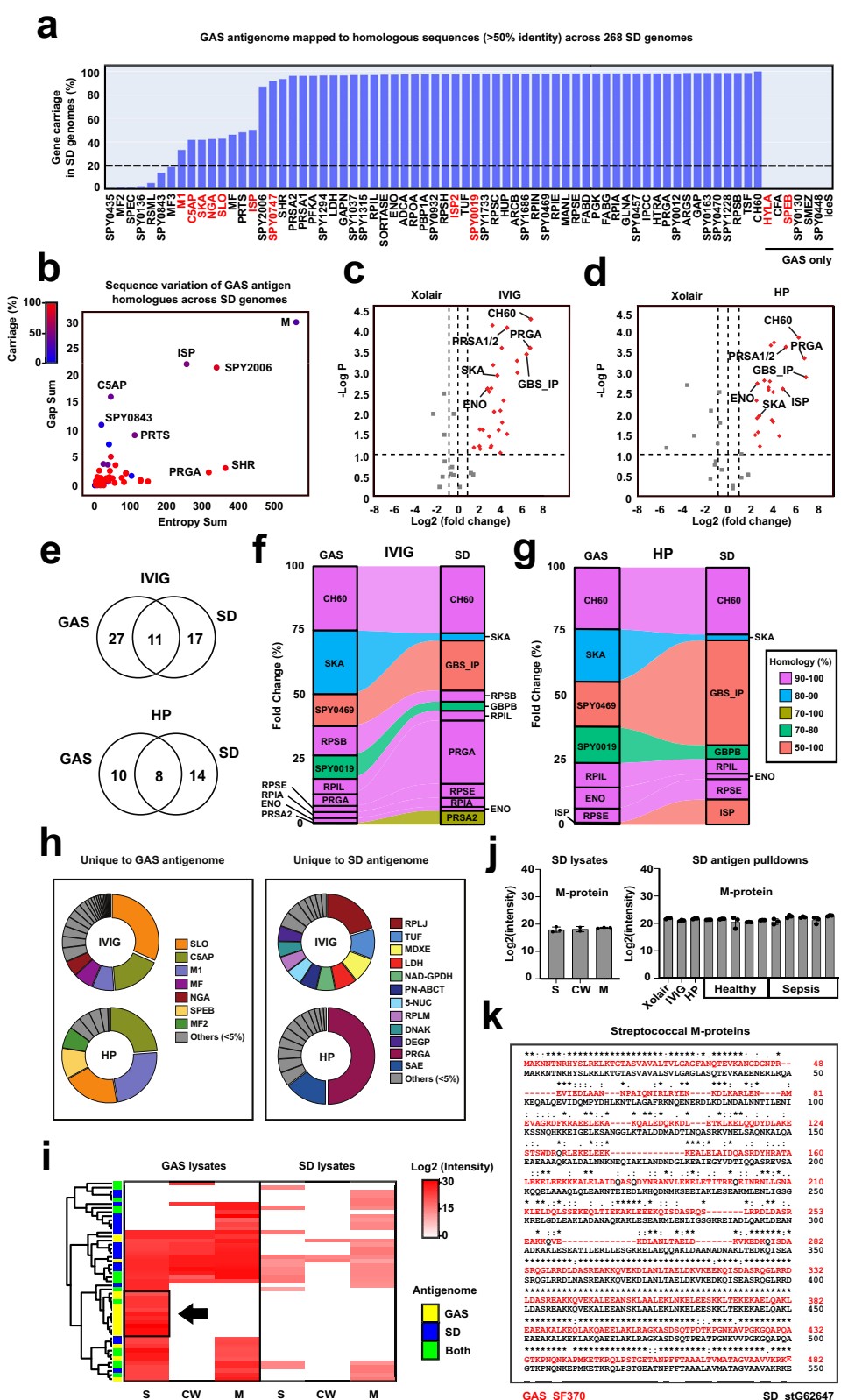

titles (M1 > C5AP > PRGA) (Fig. 5c) or glycosylation, as glycoproteomic analysis of the antigen-specific IgG ruled out major differences in the Fc glycan profiles (Supplementary data 9). However, the LC-MS/MS quantification of the affinity-purified IgG subclasses using proteotypic Fc peptides showed that more IgG of each subclass was pulled down using M1 as a bait, compared to C5AP and PRGA (Fig. 5d). The subclass distribution was also skewed and M1-antibodies were more enriched in the IgG2 and IgG3 subclasses, compared to antibodies recognizing the two other antigens.

To determine the impact of the IgG subclass of anti-M1 antibodies to trigger FcγR-signaling, we took advantage of the monoclonal mAb25 that specifically binds to the M1-protein with high affinity[34,35]. This monoclonal antibody allowed us to exclude potential confounding factors, such as the relative contribution of mixed subclasses and

**Fig. 3 | Comparative analysis of the GAS and SD antigenome. a** 72 GAS antigens mapped to homologous sequences with >50% identity across 268 different SD genomes. The 11 GAS antigens commonly identified across all individuals (Fig. 2D) are highlighted in red. **b** Sequence variation plot of the GAS antigen homologues based on the analysis of gap frequency, entropy, and gene carriage of each protein across SD genomes. Residues with high conservation have low entropy whereas residues with low conservation have high entropy. Gaps indicate insertion and deletion in sequences. **c** Volcano plot displaying the significant SD antigens recognized by IVIG and (**d**) pooled human plasma (HP). Statistical significance was determined using a both side *t*-test with an FDR of 0.05 to correct for multiple comparisons. **e** Venn diagram depicting unique and common GAS and SD antigens recognized by IVIG and HP. **f** Alluvial plots representing the fold change and

sequence homology of antigens significantly enriched from both GAS and SD lysates using IVIG and (**g**) HP. **h** Pie chart of antigens significantly enriched from either GAS or SD lysates using IVIG and HP. The area of the pie is the average intensity of triplicates expressed as a percentage of the total intensity of all antigens. **i** Differential expression of the log2 intensity of unique and common antigens of GAS or SD antigenome across secreted (S), cell wall (CW), and membrane (M) fractions of the GAS and SD lysates. (**j**) Bar plots representing log2 intensity of SD M protein across the lysates and pulldown using Xolair, IVIG, HP, plasma from healthy (*n* = 5) and convalescent plasma (*n* = 5) from sepsis individuals. Bars represent mean values and error bars represent standard deviations. Each sample was analyzed as three independent pulldowns. **k** Sequence alignment of M proteins from GAS_SF370 and SD_stG62647. Source data are provided as a Source Data file.

different epitope binding patterns of the M1-specific IgG in IVIG[36]. We tested the extent to which subclass-specific versions of mAb25 could trigger FcγR signaling. Notably, whereas mAb25 in IgG1 or IgG4 scaffolds displayed almost no measurable FcγR receptor activation, only modest CD32 activation was observed when using IgG2. However, switching to IgG3 resulted in robust induction of both CD32 and CD16 signaling (Fig. 5e, f). To rule out the possibility that the activity assay itself might have an inherent bias towards a particular IgG subclass, we used a different system in which the SARS-CoV-2 spike protein was probed with the IgG1 and IgG3 versions of the monoclonal antibody mAb81, known for triggering effective phagocytosis[37]. As expected, we found no evidence of such experimental bias (Supplementary Fig. 7). Additionally, phagocytosis assays using whole bacteria further confirmed that the IgG3 version of mAb25 promoted both binding and bacterial internalization to a higher extent compared to IgG1 (Supplementary Fig. 8).

Finally, we investigated the capacity of anti-M1 antibodies isolated from healthy and septic individuals, regardless of pathogen, to trigger CD32 and CD16 activation. Interestingly, we observed that anti-M1 antibodies from septic patients induced significantly higher CD32 activity compared to healthy donors (Fig. 5g). A similar tendency was observed for CD16 (Fig. 5h). This enhanced ability was associated with significantly higher levels of anti-M1 IgG3 antibodies in sepsis compared to healthy controls (Fig. 5i). No difference was observed for the other subclasses (Fig. 5j–l). Taken together, results from the systems serology workflow indicate that the IgG subclass, and in particular IgG3, has a major impact on the capacity of anti-M1 antibodies to trigger immune signaling.

## Discussion

In this study, we coupled MS-based systems antigenomics and systems serology to characterize polyclonal antibody responses directed against bacterial pathogens in a reproducible, high-throughput, and flexible manner, directly in human samples. The approach facilitates the identification of the antigen and epitope repertoires targeted by circulating antibodies, as well as the subclass distribution, Fc glycosylation pattern, and capacity to trigger immune signaling. To demonstrate proof-of-concept, we applied the workflow to quantify GAS-specific IgG circulating in adult human plasma since GAS remains a significant source of morbidity and mortality worldwide. A roadmap towards a GAS vaccine has recently been outlined by the World Health Organization (WHO)[38]. However, major challenges remain, including a poor understanding of the immune response against GAS and the lack of reliable immune correlates of infection and protection. Notably, the incidence of GAS infections is high in school-age children but typically declines throughout life, which suggests the buildup of protective immunity during the lifetime of an individual[28].

Systematic attempts to define protective GAS antigens have previously relied either on low-throughput approaches or on advanced protein arrays and surface display technologies[7,10,11,39]. Here, we generated libraries of bacterial proteins through basic biochemical fractionation, a strategy that is flexible, cost-effective, and amenable to

biochemical laboratories with access to standard bacterial growth facilities and equipment. Cellular fractionation also allows querying proteins in their native form and in their relevant and immunologically accessible compartments, such as membranes and cell walls. In addition, the workflow is flexible and fully automated, can be adapted to other bacteria, and can be exploited to query different growing conditions and cohorts.

Our data confirm that all adult donors and patients included in this study have circulating IgG against the GAS antigenome, a subset of streptococcal antigens that are genomically conserved across GAS isolates. This antigenome covers a wide range of bacterial proteins, including many virulence factors. Notably, we identified a common set of 11 antigens that were consistently targeted by circulating IgG across healthy individuals and patients with GAS bacteraemia. Many of these antigens, such as M1 and C5AP, have also been identified in previous studies using different experimental approaches. These repeated findings, now validated across studies performed in multiple countries, using distinct IVIG and patient sample cohorts, strongly reinforce the notion that these antigens are truly immunodominant, highlighting their potential relevance as vaccine candidates or as biomarkers for GAS exposure.

Interestingly, a typical GAS genome encodes ~1800 proteins[40,41] but the size of the antigenome is on average 30 antigens/individual, raising the question why some proteins are more frequently targeted than others. One possibility is that these virulence factors are produced in higher amounts during infection, which in turn could result in greater accessibility to immune and antigen-presenting cells. In addition, technical factors might also contribute to the apparently small size of the antigenome. Since our strategy relies on fractionated protein pools extracted from growing bacteria, the protein composition might change with different bacterial strains and culture conditions. Indeed, extending our analysis from the SF370 to two other GAS serotypes (AP1 and M49) resulted in the identification of additional antigens, including serotype-restricted proteins.

In this study, we also performed antigenome analysis on SD, a gram-positive bacterium known to colonize similar ecological niches and to trigger common disease manifestations as GAS[29]. This analysis led to the identification of previously unknown SD antigens recognized by circulating antibodies in human plasma. To our knowledge, this is the first comprehensive catalog of the antigenic repertoire of SD-specific antibodies naturally circulating in human plasma. Interestingly, the data also suggested potential immunological overlap between the GAS and SD antigenomes, based on the presence of homologous antigenic sequences across species. This overlap is most likely due to similarities in genetic composition resulting from frequent horizontal gene transfer (HGT), which increases the possibility of immunological cross-reactions[30,31]. There is some evidence that antibodies elicited against one bacterial pathogen can provide protection against another pathogen due to cross-reactivity[42–44]. Whether similar mechanisms apply to GAS and SD, or even other streptococcal species, remains to be investigated. This is particularly important for vaccine design efforts, as targeting shared antigens might result in

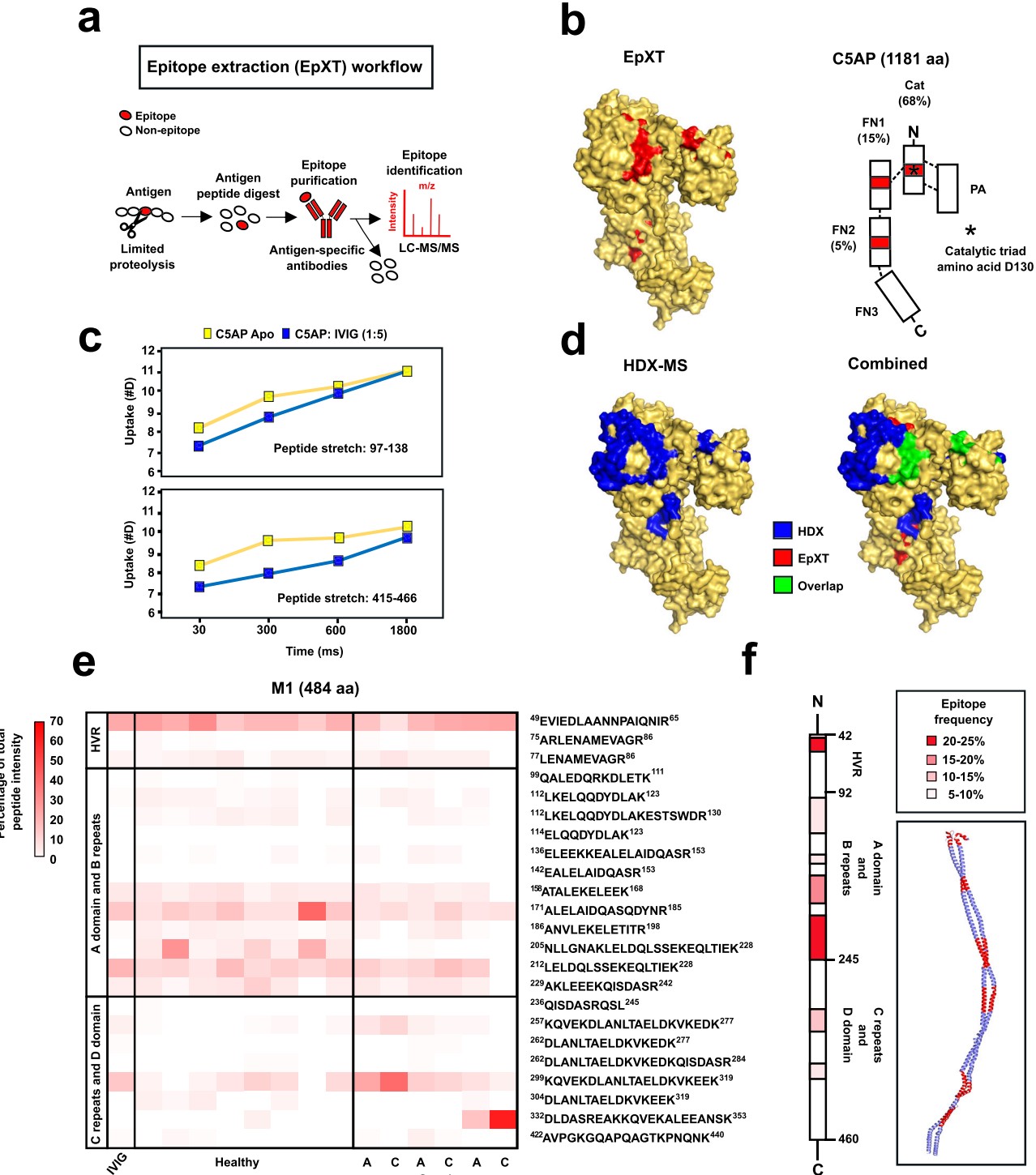

**Fig. 4 | Epitope mapping of GAS antigens. a** Schematic representation of the epitope extraction workflow (EpXT) to identify epitopes on recombinantly expressed antigens. **b** Identified peptides (marked red) by EpXT mapped onto the crystal structure of C5AP (left) and their relative intensity (%) is shown on the C5AP cartoon (right). **c** Deuterium uptake plots for two peptide stretches of C5AP alone (yellow) and when incubated with IVIG (blue). **d** Identified peptides (marked blue) by HDX-MS (left) and overlapping epitopes identified by both HDX-MS and EpXT mapped onto the crystal structure of C5AP (right). **e** Heatmap of M1 peptide intensities across IVIG, healthy, and sepsis individuals. **f** Consensus epitope landscape across all individuals with more than 5% epitope frequency are displayed in the M1 cartoon and the M1 model.

significant antimicrobial and epidemiological effects on both pathogen populations. Finally, our data also revealed that the antigenome of patients with GAS bacteremia differs from the antigenome of healthy uninfected individuals, suggesting that it might be dynamically regulated by the immune status. However, antigenome analysis of SD failed to discriminate healthy donors from SD-septic patients, suggesting

that pathogen-specific factors might also be important. Future studies with larger and more defined clinical cohorts are needed to unravel the molecular basis for these differences.

In addition to antigenomic analysis, our methodology enabled the mapping of the epitope landscape of selected GAS antigens. Interestingly, antibody recognition was invariably associated with defined

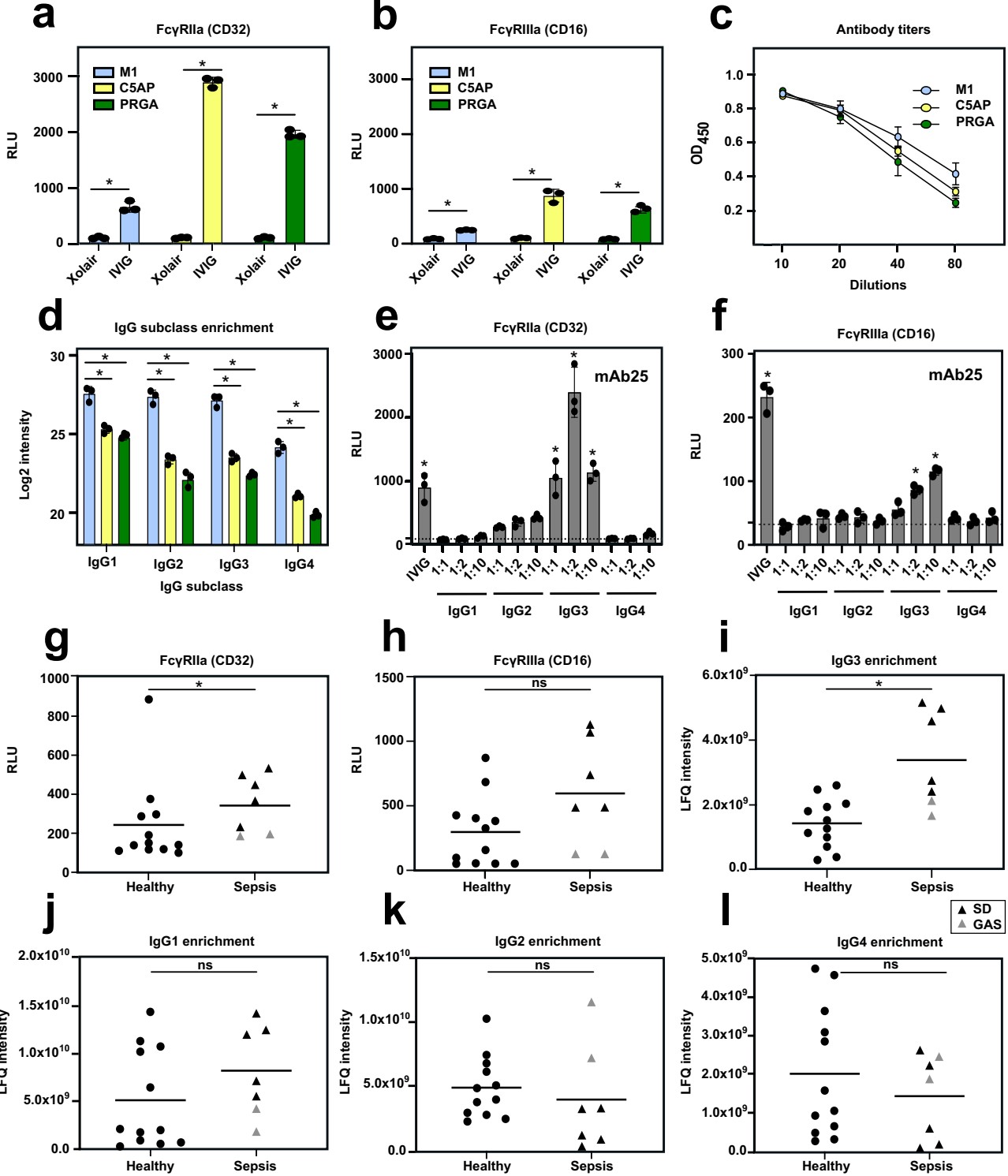

**Fig. 5 | GAS-specific antibodies trigger immune signaling in an antigen- and receptor-specific manner. a** FcγRIIa (CD32) and (**b**) FcγRIIIa (CD16) activity assay of M1, C5AP, and PRGA specific antibodies present in IVIG. **c** Antibody titers in IVIG against M1, C5AP, and PRGA. **d** Subclass enrichment profiles of antigen-specific IgG in IVIG. **e** FcγRIIa (CD32) and (**f**) FcγRIIIa (CD16) activity assay of mAb25 in IgG1-4 scaffolds. Dotted lines indicate the background values. Statistical significance was assessed by either one-way or two-way ANOVA with correction for multiple comparisons, $*p < 0.05$. The results are the average of experiments done in triplicates. Bars represent mean values and error bars represent standard deviations. **g** FcγRIIa (CD32) and (**h**) FcγRIIIa (CD16) activity assay of M1-specific antibodies across healthy and sepsis individuals. Quantification of M1-specific (**i**) IgG3, (**j**) IgG1, (**k**) IgG2, and (**l**) IgG4 enriched from plasma derived from healthy ($n = 12$) and sepsis individuals ($n = 7$) based on the LFQ intensity of the most intense proteotypic peptide of each IgG subclass. Statistical significance was assessed by a two-sided Mann–Whitney unpaired *t*-Test, $*p < 0.05$. The results are the average of experiments done in triplicates and bars represent mean values. Source data are provided as a Source Data file.

antigenic sites or immunodominant regions[45]. For example, the HVR and the variable region of the M-protein were found to be major interaction sites for GAS antibodies. The variability of the HVR is thought to be the result of selective pressure on the bacteria to escape the immune response since type-specific antibodies are protective against infections[46,47]. However, previous animal studies indicated that the HVR might be only weakly immunogenic[48]. One possible explanation for this discrepancy is that GAS infection in humans is accompanied by a robust induction of the immune response, which might create an appropriate environment for selection of B-cell clones targeting the HVR. These conditions might not be completely phenocopied by immunization studies using laboratory animals.

The affinity of FcγRs for IgG varies with the Fc structure, in particular the subclass and/or glycosylation[2]. Here we show that GAS-antibodies can trigger immune signaling through engagement of multiple FcγRs, in an antigen- and Fc receptor-specific manner. For M1-specific antibodies, septic individuals exhibited a higher capacity to induce immune signaling compared to healthy controls, which correlated with increased levels of antigen-specific IgG3. This enhanced opsonophagocytic ability of anti-M1 specific antibodies of the IgG3 subclass is most likely related to the flexibility of its larger hinge region compared to other subclasses, as previously demonstrated[35,37,49–53]. Although the basis for this shift towards the IgG3 subclass remains unclear, one potential explanation might be that immune activation during streptococcal sepsis might trigger expansion/differentiation of IgG3-specific B-cell clones. Interestingly, previous studies have found that immune responses against GAS in adults are dominated by higher IgG3 and TNF-α levels compared to children, which correlates with the buildup of a protective immunity throughout life[54]. At the same time, elevated anti-M IgG3 levels have recently been observed in patients suffering from acute rheumatic fever[55]. These apparently conflicting observations highlight the need for better immune correlates of GAS infection, protection, and susceptibility to autoimmune sequelae. Nevertheless, a key contribution of this study lies in its ability to link antigen specificity to functional antibody properties, which is poorly understood in the context of GAS immunity. This is particularly evident in this analysis of antibody responses against the M1 protein, where we identified that M1-specific antibodies target immunodominant regions, including the hypervariable and variable regions while exhibiting subclass-specific functional properties. For example, anti-M1 antibodies from septic patients displayed an increased proportion of IgG3, which was associated with enhanced Fcγ-receptor engagement and opsonophagocytic activity. These findings highlight the strength of our integrated workflow in coupling antigenic targets to immune functions, to start addressing this long-standing gap in GAS immunology.

Finally, FcγR signalling is also dependent on the capacity of the antibody to induce immune complexation with the antigen, the nature of the antigen itself, and the geometry of antigen-antibody interactions. In this context, it is important to note that the signaling assay used in this study relies on coating antigens onto a plate, which may restrict antibody binding and flexibility, potentially affecting receptor dimerization. Exploring alternative systems, such as immobilizing antigens on beads or similar platforms to circumvent potential steric hindrances, could provide additional insights. Similarly, complementing the signaling assays with phagocytosis assays using other cell lines that, unlike the THP1 cell line, express complement receptors could provide a more comprehensive understanding of Fc function, especially for bacterial antibodies where opsonophagocytic killing (OPK) is likely critical for protection. Lastly, while this study provides significant insights into the antigenic and functional properties of GAS-specific IgG, several limitations should be acknowledged. The use of a limited number of GAS strains in this study may not fully capture the genomic and antigenic diversity of the pathogen, and the small clinical cohort may restrict the generalizability of some findings. Additionally,

the focus on IgG responses excludes other antibody isotypes, such as mucosal IgA, which is highly relevant to the immune response against GAS. Nonetheless, the robustness and flexibility of our integrated methodology provide a powerful framework to dissect antibody responses against complex pathogens. Future studies expanding strain diversity and clinical cohorts will build on these findings to further advance our understanding of streptococcal immunity and accelerate vaccine development.

## Methods

### Patient enrolment and sample collection
The sampling of patients with bacteremia was approved by the regional ethics committee of Lund University, (2016/939, with amendment, 2018/828). Oral and written consents were obtained from included participants. During 2018–2020, four patients with GAS bacteraemia and during 2017–2018 five patients with *S. dysgalactiae* bacteraemia in Region Skåne, Sweden, were enrolled in the study. Acute blood samples were collected within five days after hospital admission, and convalescent blood samples were collected after 4–6 weeks. Information on the included patients is given by de Neergaard et al. [27] and Bläckberg et al. [56]. Sampling of healthy donor blood was approved by the regional ethics review board in Lund, Sweden (approval 2015/801). Citrated blood samples were collected from healthy donors without any sign of ongoing infection and otherwise in good health. Platelet-poor plasma was prepared by centrifugation at $2000 \times g$ for 10 min and stored at $-80\,°C$ until use.

### Biochemical fractionation of GAS proteins
A single colony of the GAS strains SF370, AP1, M49, or SD strain stG62647 was precultured in Todd-Hewitt broth supplemented with 0.6% yeast extract (THY) at $37\,°C$ and 5% $CO_2$ for 16–18 h ($OD_{620\,nm}$ = 0.8) and then the bacteria were sub-cultured in either protein reduced THY broth or regular THY broth. The protein-reduced THY broth was prepared by passing THY broth through a 0.22-μm-pore-size-filter and then filtered using a 10-kDa molecular mass cut-off. For the secreted fractions, bacteria grown in protein-reduced THY broth at $37\,°C$ and 5% $CO_2$ till the mid-logarithmic phase ($OD_{620\,nm}$ = 0.4–0.5) were harvested at $3000 \times g$ for 15 min at $4\,°C$ and the culture supernatant was filtered using a 0.22-μm-pore-size-filter unit. The filtered supernatant was buffer exchanged to PBS and concentrated using an amicon ultracel 10 kDa molecular weight cutoff centrifugal filtration unit (Millipore) at $4000 \times g$ for 15 min (min) and stored in $-20\,°C$ until further use.

For the cell wall and membrane fractions, bacteria were sub-cultured in regular THY broth at $37\,°C$ and 5% $CO_2$ to mid-logarithmic phase ($OD_{620\,nm}$ = 0.4–0.5) and the cells were harvested at $3000 \times g$ for 15 min at $4\,°C$. The bacterial pellets were kept on ice for a brief period of 5 min followed by resuspension in 5 ml chilled TES buffer (50 mM Tris-HCl, 1 mM EDTA, 20% sucrose (w/v) sucrose, pH 8.0) containing 1 mM phenylmethylsulfonyl fluoride (PMSF, Roche) at $3500 \times g$ for 20 min at $4\,°C$. For bacterial cell wall lysis, 1.15 ml of ice-cold mutanolysin mix 1 ml TES buffer, 100 μl lysozyme (100 mg/ml in TES), 50 μl mutanolysin (Sigma–Aldrich, 5000 U/ml in 0.1 M $K_2HPO_4$, pH 6.2) was added to the cells and incubated for 2 hr at $37\,°C$ at 200 rpm shaking. Cells were then centrifuged at $14,000 \times g$ for 5 min and the resulting supernatant was stored at $-20\,°C$ until further use.

To isolate membrane proteins, the cell pellets were washed twice in 1 ml HEPES buffer at $3500 \times g$ for 5 min and the cells were then resuspended in 1% HEPES. One microgram sequencing-grade modified trypsin (Promega) was added to cells for 60 min at $37\,°C$ at 500 rpm to shave off the membrane proteins and the reaction was stopped by incubating the cells on ice for 2 min before centrifuging them at $1000 \times g$ for 15 min at $4\,°C$. The supernatant

containing the membrane proteins was then collected and stored at −20 °C. Cell pellets were further treated with RIPA lysis buffer for 15 min and centrifuged at $3500 \times g$ for 5 min to collect the intracellular fractions.

## Affinity purification of bacterial antigens

IgGs from different sources were purified in a 96-well plate (Greiner) using the Protein G AssayMAP Bravo (Agilent) system, according to the manufacturer's instructions. Briefly, 100 μg of IVIG, 100 μg of Xolair, and 10 μl (-100 μg IgG) of human plasma were diluted with PBS to a final volume of 100 μl and then applied to pre-equilibrated Protein G columns. Columns were washed with PBS, before applying a pool of 100 μg secreted, 100 μg cell wall, and 100 μg membrane fractions. The antigen-antibody complex was then eluted in 0.1 M glycine (pH = 2) and the final pH was neutralized with 1 M Tris, and saved until further use. The proteins were denatured using 8 M urea solution and 5 mM Tris(2-carboxyethyl) phosphine hydrochloride (TCEP) was then added for 60 min at 37 °C to reduce the disulfide bonds followed by alkylation with 10 mM iodoacetamide in the dark at room temperature for 30 min. One hundred millimiter ammonium bicarbonate was added followed by the addition of trypsin (1:100) for protein digestion at 37 °C for 18 h. The activity of trypsin was inhibited by dropping the pH to 2–3 by the addition of 20% trifluoroacetic acid (TFA, Sigma). The samples were loaded on Evosep tips to separate the digested peptides using nanoflow reversed-phase chromatography on an Evosep One liquid chromatography (LC) system (Evosep One) and analyzed on timsTOF Pro mass spectrometer (Bruker Daltonics).

## Antigen-specific IgG pulldowns

Antigen-specific IgG was purified from IVIG and human plasma in a 96-well plate setup according to the manufacturer's instructions. Roughly 100 μl of human plasma was used for IgG enrichment using the Protein G AssayMAP Bravo (Agilent) technology, buffer exchanged using 50 K centrifugal filters (Amicon Ultra-0.5 ml, Merck) for 10 min at $14,000 \times g$, and finally resuspended in 100 μl of PBS. For the antigen-specific pulldowns, 20 μg of recombinantly expressed M1, C5AP, and PRGA with streptavidin tag were immobilized on pre-equilibrated AssayMAP Streptavidin columns (Agilent Technologies). Columns were washed with PBS and then IVIG (100 μg) or enriched plasma IgG were applied, followed by PBS wash. Elution was done using 100 μl of 0.1 M glycine (pH = 2) and the final pH was neutralized with 20 μl of 1 M Tris. Samples were reduced with 5 mM Tris(2-carboxyethyl) phosphine hydrochloride (TCEP), alkylated with 10 mM iodoacetamide, and digested with trypsin (1:100) at 37 °C for 18 h, and the digestion was stopped using 20% TFA (Sigma) to pH 2–3. Peptide clean-up was performed using AssayMAP C18 columns (Agilent Technologies) according to manufacturer's protocol. Samples were dried using vacuum concentrator (Eppendorf) and resuspended in 20 μl 0.1% formic acid (FA, Fisher Chemical) followed by a brief sonication for 5 min before analyzing on a Q Exactive HF-X mass spectrometer (Thermo Scientific).

## Epitope extraction (EpXT)

To benchmark the EpXT workflow Pierce Protein G magnetic beads (Thermo Scientific) were used. For IgG enrichment, 50 μl of protein G beads were washed with PBS, before 100 μg of IVIG diluted with PBS to a final volume of 100 μl was added and incubated for 1 h followed by PBS wash. Ten micrograms of recombinant C5AP and PRGA were trypsinized with 1 μg of trypsin for 15 min at 37 °C and the trypsin activity was inhibited by incubating at 100 °C for 5 min. The peptide digest was then incubated with protein G enriched IgG for 1 h and then washed with PBS before eluting with 100 μl of 0.1 M glycine (pH = 2) and the pH was finally neutralized with 1 M Tris. Peptide clean-up was performed on Evosep columns as mentioned above before analyzing on a timsTOF Pro mass spectrometer (Bruker). As a control, a mixture

of the digested recombinant proteins was loaded on the Protein G columns without reactive polyclonal antibodies, to assess non-specific binding to the affinity columns. Enriched peptides were required to have a significant fold change increase over this background (>2-fold).

For M1 EpXT analysis IgGs from IVIG and human plasma were purified in a 96-well plate (Greiner) using the Protein G AssayMAP Bravo (Agilent) system. One hundred micrograms of IVIG and 10 μl of human plasma (-100 μg IgG) were diluted with PBS to a final volume of 100 μl and then applied to pre-equilibrated Protein G columns. Columns were washed with PBS, before applying the M1 peptide digest. The M1 peptide digest was prepared by incubating 10 μg of M1. One microgram trypsin at 37 °C for 15 min followed by a brief incubation at 100 °C for 5 min. After PBS wash, the M1 peptide-antibody complex was then eluted in 0.1 M glycine (pH = 2) and the final pH was neutralized with 1 M Tris. Peptides were cleaned up on Evosep columns according to the manufacturer's instructions and analyzed on a timsTOF Pro mass spectrometer (Bruker Daltonics).

## FcγR-luciferase reporter cell assay

Jurkat-Lucia NFAT-CD16 (FcγRIIa) and CD32 (FcγRIIIa) cells (InvivoGen) were used to probe the ability of antigen-specific IgG to trigger antibody-dependent cellular cytotoxicity (ADCC) and antibody-dependent cell-mediated phagocytosis (ADCP). Nunclon delta surface plates (Thermo Scientific) were coated with 0.5 μg of M1, C5AP, PRGA, and SARS-CoV-2 spike protein overnight at 4 °C, followed by PBS wash. 100 μg of IVIG, 100 μg of Xolair (Omalizumab, 150 mg, Novartis), 10 μl of human plasma (-100 μg IgG), various amounts of the mAb25 (0.5, 1.0 and 5 μg)[35] or mAb81 (0.5, 1.0 and 5 μg)[37] expressed in different IgG subclasses were diluted to a final volume of 100 μl with PBS and incubated for 1 h at 37 °C. After PBS wash, 200 μl of CD16 and CD32 cells (100,000 cells/100 μl) in IMDM with 10% heat-inactivated fetal bovine serum (FBS) and Pen-Strep (100 U/ml–100 μg/ml) were respectively added and incubated at 37 °C for 6 h. After brief centrifugation for 10 min at $150 \times g$, 20 μl of the supernatant was added to 50 μl of QUANTI-Luc (InvivoGen) in opaque microtiter plates, and the luciferase activity was measured in a luminometer.

## IgG immunoblotting

Secreted, cell wall and membrane GAS protein fractions were separated on SDS-PAGE (Criterion TGX Gels, 4–20% precast gels, Bio-Rad) and proteins were transferred to PVDF membranes using the trans-blot turbo transfer system (Bio-Rad) according to the manufacturer's instructions. The membranes were blocked with 5% bovine serum albumin (BSA) in PBST (PBS + 0.05% Tween 20) for 1 h at 37 °C, followed by incubation with 100 μg commercially purchased IVIG (Octagam 100 mg/ml, Octapharma) and commercially purchased pooled human plasma (Innovative research) (1:10 i.e.,100 μg IgG) overnight at 4 °C. After washing, the membranes were incubated with Protein G-HRP conjugate (1:3000, Bio-Rad) for 1 h at 37 °C. The membranes were developed using clarity western ECL substrate (Bio-Rad) and visualized in the ChemiDoc MP Imaging System (Bio-Rad).

## Enzyme-linked immunosorbent (ELISA) assay

To measure GAS-specific antibodies, 96 well Nunc microtiter plates (Thermo Scientific) were coated with 0.5 μg of recombinant M1, C5AP, and PRGA overnight at 4 °C followed by PBST (PBS + 0.05% Tween 20) wash. Plates were blocked with 2% BSA (100 μl/well) in PBST for 30 min at 37 °C. After washing with PBST, 100 μg of IVIG and plasma (1:10, 100 μg IgG) were added in dilution series in triplicates and incubated at 37 °C for 1 h and then washed with PBST. 100 μl/well of Protein G-HRP conjugate (1:3000, Bio-Rad) in PBS was added and incubated for 1 h at 37 °C and then washed with PBST. The reaction was developed using 100 μl/well ABTS (20 ml Sodium Citrate pH 4.5 + 1 ml ABTS + 0.4 ml H2O2) for 30 min and the OD was measured at 450 nm.

## Phagocytosis assay

Phagocytosis assays were done with THP-1 cell line (Sigma-Aldrich) and heat-killed SF370 bacteria, stained with Oregon green 488 x succini-midyl 696 ester (Thermofisher) and Cypher5e (Fisher Scientific), as done previously[33,55]. $1 \times 105$ cells were used, while bacterial quantity varied (Multiplicity of Prey, MOP: 50, 25, and 12.5). Bacteria were opsonized in a volume of 100 µl of sodium media (pH adjusted to 7.3 with NaOH; 5.6 mM glucose, 10.8 mM KCl, 127 mM NaCl, 2.4 mM $KH_2PO4$, 1.6 mM $MgSO4$, 10 mM HEPES, 1.8 mM $CaCl_2$). The bacteria were opsonized with either 1 mg/mL of IVIG, 1 mg/mL of purified IgG from pooled human plasma, 10 µg/ml of mAB25, or 20 µg/ml Xolair. The opsonization occurred for 30 min at 37 °C on a shaking heat block on a 96-well plate. After 30 min, $1 \times 105$ THP-1 cells suspended in sodium media were added in a volume of 50 µl. The cells were left to phagocytose the bacteria for 30 min at 37 °C on a shaking heat-block. Subsequently, the 96-well plate was put on ice for 20 min to stop phagocytosis. After 20 min, the plate was analyzed in a Beckman Colter Cytoflex flow cytometer. A gate for THP1 cells was set up based on their forward and side scatter and duplicate events were excluded by a single-cell gate with side scatter area and side scatter height. The gate for internalization and association was set with negative control of cells only. The analysis stopped after 5000 events were captured in the THP-1-gate. The data was analyzed in the program Flowjo by setting similar gates as described above. The Flowjo-processed data was further analyzed in GraphPad Prism.

## LC-MS/MS proteome analysis

Peptide analysis using data-dependent mass spectrometry (DDA-MS) was performed on a Q Exactive HFX instrument (Thermo Scientific) connected to an Easy-nLC 1200 system (Thermo Scientific). An Easy-Spray column (50 cm, column temperature of 45 °C, Thermo Scientific) operated at a maximum pressure of $8 \times 107$ Pa separated the peptides, and a linear gradient of 4–45% acetonitrile in aqueous 0.1% formic acid was run for 65 min. One full MS scan (resolution of 60,000 for a mass range of 390 to 1210, automatic gain control = 3e6) was followed by MS/MS scans (resolution of 15,000, automatic gain control = 1e5) of the 15 most abundant signals. 2 m/z isolation width was set for precursor ions and higher-energy collisional-induced dissociation (HCD) at a normalized collision energy of 30 was used for fragmentation. For peptide analysis on timsTOF Pro, a 30 SPD method (gradient length = 44 min) was used for the separation using an $8 \times 150$ m Evosep column packed with 1.5 m ReproSil-Pur C18-AQ particles. A captive source coupled to Evosep One was mounted on the timsTOF Pro mass spectrometer (Bruker Daltonics) which was operated in DDA PASEF mode with 10 PASEF scans per acquisition cycle with accumulation and ramp times of 100 ms each. The target value was set to 20,000 dynamic exclusion was set to 0.4 min and singly charged precursors were excluded. The isolation width was 2 Th for m/z < 700 and 3 Th for m/z > 800.

## Glycoproteomics analysis

Purified IgG glycopeptides were analyzed on a Q Exactive HF-X mass spectrometer (Thermo Fisher Scientific) connected to an EASY-nLC 1200 ultra-HPLC system (Thermo Fisher Scientific). Peptides were trapped on precolumn (PepMap100 C18 3 µm; 75 × 2 cm; Thermo Fisher Scientific) and separated on an EASY-Spray column (Thermo Fisher Scientific). Mobile phases of solvent A (0.1% formic acid), and solvent B (0.1% formic acid, 80% acetonitrile) were used to run a linear gradient from 4 to 45% over 60 min. MS scans were acquired in data-dependent mode with the following settings, 60,000 resolution @ m/z 400, scan range m/z 600–1800, maximum injection time of 200 ms, stepped normalized collision energy (SNCE) of 15 and 35%, isolation window of 3.0 m/z, data-dependent HCD-MS/MS was performed for the ten most intense precursor ions.

## Hydrogen-deuterium exchange mass spectrometry (HDX-MS)

The HDX-MS analysis was made using automated sample preparation on a LEAP H/D-X PAL™ platform interfaced to an LC-MS system, comprising an Ultimate 3000 micro-LC coupled to an Orbitrap Q Exactive Plus MS. HDX was performed on 1.2 mg/ml C5AP and IVIG (8 mg/mL), in 1X PBS, at a ratio of 1:2 and 1:5 in one continuous run, with runs of the apo state made in between the interaction runs. In total, 4 replicate runs were made for the apo state and triplicates for the antibody interaction states. Five microliter HDX samples were diluted with 25 µl 20 mM PBS, pH 7,4, or HDX labeling buffer of the same composition prepared in D2O, pH(read) 7.0. The HDX labeling was carried out for t = 0, 30, 300, 600, and 1800 s at 4 °C. The labeling reaction was quenched by dilution of 30 µl labeled sample with 30 µl of 1% TFA (Sigma), 0.4 M TCEP (Sigma), 4 M Urea (Sigma), pH 2.5 at 1 °C. Sixty microliters of the quenched sample were directly injected and subjected to online pepsin digestion at 4 °C (in-house immobilized pepsin column, 2.1 × 30 mm). The online digestion and trapping were performed for 4 min using a flow of 50 µL/min 0.1% FA (Sigma), pH 2.5. The peptides generated by pepsin digestion were subjected to on-line SPE on a PepMap300 C18 trap column (1 × 15 mm) and washed with 0.1% FA (Sigma) for 60 s. Thereafter, the trap column was switched in line with a reversed-phase analytical column, Hypersil GOLD, particle size 1.9 µm, 1 × 50 mm, and separation was performed at 1 °C using a gradient of 5–50% B over 8 min, and then from 50 to 90% B for 5 min, the mobile phases were 0.1% FA (A) and 95% acetonitrile/0.1% FA (B). Following the separation, the trap and column were equilibrated at 5% organic content, until the next injection. The needle port and sample loop were cleaned three times after each injection with mobile phase 5% methanol (MeOH)/0.1% FA, followed by 90% MeOH/0.1% FA, and a final wash of 5% MeOH/0.1% FA. After each sample and blank injection, the Pepsin column was washed by injecting 90 µL of pepsin wash solution 1% FA/4 M urea/5% MeOH. In order to minimize carry-over a full blank was run between each sample injection. Separated peptides were analyzed on a Q Exactive Plus MS, equipped with a HESI source operated at a capillary temperature of 250 °C with sheath gas 12, Aux gas 2, and sweep gas 1 (au). For HDX analysis MS full scan spectra were acquired at 70 K resolution, automatic gain control = 3e6, Max IT 200 ms, and scan range 300–2000. For identification of generated peptides separate undeuterated samples were analyzed using data-dependent MS/MS with HCD fragmentation.

## HDX-MS data analysis

PEAKS Studio X Bioinformatics Solutions Inc. (BSI, Waterloo, Canada) was used for peptide identification after pepsin digestion of undeut-erated samples. The search was done on a FASTA file with the sequence of C5AP. Search allowed a mass error tolerance of 15 ppm a fragment mass error tolerance of 0.05 Da, and fully unspecific cleavage by pepsin. Peptides identified by PEAKS with a peptide score value of log $P > 25$ and no modifications were used to generate peptide lists containing peptide sequence, charge state, and retention time for the HDX analysis. HDX data analysis and visualization were performed using HDExaminer, version 3.1.1 (Sierra Analytics Inc., Modesto, US). The analysis was made on the best charge state for each peptide, allowed only for EX2, and the two first residues of a peptide were assumed unable to hold deuteration. Due to the comparative nature of the measurements, the deuterium incorporation levels for the peptic peptides were derived from the observed relative mass difference between the deuterated and non-deuterated peptides without back-exchange correction using a fully deuterated sample[57]. As a full deuteration experiment was not made, full deuteration was set to 75% of maximum theoretical uptake. In our system, achieving 100% deuteration is not feasible due to two primary factors: a dilution with $D_2O$ of -1:10 and back-exchange occurring within the system[57,58]. To account for this, we set the maximum deuteration level to 75% and ensure that

no peptide exceeds 100% theoretical uptake[59]. While this approach may result in slightly over- or underestimation of actual uptake values the analysis is still based on relative comparisons of uptake between bound and unbound states. The presented deuteration data is the average of all high and medium confidence results. The allowed retention time window was ±0.5 min. The spectra for all time points were manually inspected.

## Proteomics data analysis

The DDA data was analyzed in MaxQuant (version 2.0.3.0). The protein database used for the searches were *Homo sapiens proteome* (UniProt proteome identifier UP000005640), GAS proteome (UniProt proteome identifier UP000000750) compiled with common contaminants. Two hundred sixty-eight genomes of *Streptococcus dysgalactiae* were retrieved from the Bacterial and Viral Bioinformatics Resource Center (BV-BRC) and quality control measures were applied, to exclude genomes classified as poor quality, plasmid, duplicated, or poorly annotated such that 173 genomes were retained for further analysis. These sequences were integrated into a local SeqRepo database, for efficient sequence retrieval and management. The 71,646 unique sequences in the database were further subjected to additional filtering to remove sequences with less than 100 amino acids and incomplete sequences containing ambiguous residues. Following these, a dataset of 58,572 high-quality sequences was subjected to redundancy reduction by clustering, using CD-HIT[60] at a 70% similarity threshold, resulting in 6164 clusters, all identified by a representative sequence which was finally treated as the SD proteome database. The enzyme was set to trypsin with up to 2 missing cleavage allowed and a mass error tolerance of 10 ppm for both precursors and fragments was allowed. Carbamidomethyl (C) modification was set as fixed modification and oxidation (M) and acetyl (Protein N-term) were set to variable modification. One percent protein false discovery rate (FDR) was allowed and match between runs was enabled. The LFQ intensities reported by MaxQuant were used for analysis. The resulting DDA data sets were analyzed in Perseus (version 1.6.15.0 and 2.0.7.0) and R studio (version 4.2.0). Both side *t*-test with an FDR of 0.05 was used for volcano plot analysis. Positive antigenome identifications required at least a 2-fold enrichment over background samples, keeping all conditions the same except for the use of a control IgG (anti-IgE, Xolair) instead of polyclonal plasma IgG.

## Glycoproteomics data analysis

Raw files were analyzed using Byonic (Protein Metrics Inc., v5.0.3), integrated as a node within Proteome Discoverer (Thermo, v2.5.0.400). The analysis was conducted against a UniProt human protein database, employing the default search parameters: trypsin as the enzyme with up to two missed tryptic cleavages per peptide, allowance for one glycan per peptide as a "rare" variable modification, mass deviation thresholds of up to 10/20 ppm for precursor and product ions, and up to one methionine oxidation (+15.994 Da) per peptide as a variable "common" modification. The search also incorporated a predefined plasma glycan database of human structures, as well as a decoy and contaminant database provided in Byonic.

## Antigenome network analysis

All statistical methods were implemented using Python (version 3.6.10). Antigen intensities across IVIG, pooled human plasma, healthy donor plasma, and acute and convalescent phase plasma were scaled and ranked. First, the antigen-by-antigen Kendall Tau measure was made for correspondence in antigen presentation. Second the sample-by-sample Kendall Tau measure was made for correspondence in sample antigen profiles. The Benjamini–Hochberg procedure was used to control for an FDR of <0.10. The significant Kendall Tau measures formed a network where nodes are defined by antigens and edges

between nodes are the Kendall Tau measure. Visualization and analysis of the network layers were conducted through Cytoscape.

## Gene carriage, entropy, and gap analysis for GAS antigenome

A sequence database was built using all *Streptococcus pyogenes* genomes available in The Bacterial and Viral Bioinformatics Resource Center (BV-BRC, as of 2023-03-28). The sequences of all the antigens were separately searched towards the database using BLASTp, and all hits covering more than 70% of the query sequence were extracted and multiple sequence alignment (MSA) was generated with MUSCLE (version 3.8.1551). The Shannon entropy and frequency of gaps were calculated to indicate the level of conservation within a group of sequences.

## Reporting summary

Further information on research design is available in the Nature Portfolio Reporting Summary linked to this article.

# Data availability

The mass spectrometry data and the HDExaminer analysis files generated in this study have been deposited in the MassIVE repository under accession code MSV000093310 [MassIVE Dataset Summary], and are publicly available. Source data are provided with this paper.

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

## Acknowledgements

J.M. is a Wallenberg academy fellow (KAW 2017.0271) and is also funded by the Swedish Research Council (Vetenskapsrådet, VR) (2019-01646 and 2018-05795), the Wallenberg Foundation (WAF grant number 2017.0271), and Alfred Österlunds Foundation. L.M. is funded by the Swedish Research Council (Vetenskapsrådet, VR) (VR-2020-02419) and Alfred Österlunds Foundation. This work was supported by generous funding from NIGMS (R35 GM119850 to N.E.L.) and the Novo Nordisk Foundation (NNF20SA0066621 to N.E.L.). We highly acknowledge Lund Protein Production Platform (LP3) for the recombinant protein expression and Di Tang for providing the recombinant C5AP protein. We thank Ganna Petruk for collecting blood from healthy donors. Support from the Swedish National Infrastructure for Biological Mass Spectrometry (BioMS), the SciLifeLab, and Integrated Structural Biology platform is gratefully acknowledged.

## Author contributions

A.G.T., S.C., and J.M. conceived the project. A.G.T., S.C, A.I., B.O., S.E., and S.K. conducted experiments. A.G.T., S.C., E.H., A.I., B.O., S.E., J.S., N.E.L., P.N., L.M., and J.M. analyzed and interpreted data. A.B., M.R., and P.N. provided clinical samples and antibody reagents. A.G.T., S.C., and J.M. wrote the paper with significant input from all co-authors.

## Funding

## Competing interests

The authors declare no competing interests.

## Additional information

¹Division of Infection Medicine, Department of Clinical Sciences, Faculty of Medicine, Lund University, Lund, Sweden. ²Bioinformatics and Systems Biology Graduate Program, University of California, San Diego, CA, USA. ³Departments of Pediatrics and Bioengineering, University of California, San Diego, CA, USA. ⁴BioMS, Lund, Sweden. ⁵These authors contributed equally: Alejandro Gomez Toledo, Sounak Chowdhury. ✉e-mail: johan.malmstrom@med.lu.se

