## [Peer Review File · Nature Communications]

Dissecting the properties of circulating IgG against streptococcal pathogens through a combined systems antigenomics-serology workflow

Corresponding Author: Professor Johan Malmström

Version 0:

Reviewer comments:

Reviewer #1

(Remarks to the Author)

The authors have used what they term an integrative mass spectrometry-based approach which combines systems antigenomics and systems serology to characterize human antibodies to Group A Streptococcus (GAS). Unsurprisingly, both healthy and GAS-infected individuals have circulating IgG against genomically conserved streptococcal proteins. Again, as might be expected, there was inter-individual variation in titers, subclass distributions, and Fc signalling capacity. There was cross-reactivity with Streptococcus dysgalactiae. The novelty of this manuscript is in the integration of the technology.

Further comments

1) ABSTRACT:

- a) There is insufficient detail for the reader to determine the precise nature of the technology or the results.
- b) It is stated in the significance statement that "The method is cost effective, flexible, and adaptable" but this is not built on evidence provided in the manuscript.

2) INTRODUCTION:

- a) This provides a synopsis of IgG biology at a level not required for a manuscript of this nature
- b) Systems antigenomics and serology are reviewed at a very high level and not detail is provided.

2) METHODS & RESULTS:

- a) An overall workflow and data integration image would have been useful.
- b) The analysis of employed commercial sera from an undisclosed number of healthy donors and commercial IVIG, and from 4 patients with GAS bacteraemia and 5 patients with S. dysgalactiae bacteraemia (details not provided). Ten donors were used to determine inter-individual variation but these were incompletely described.
- c) It appears that both sera and citrated plasma were used but no data is providing as to the impact of the collection methods.
- d) Most of the analysis was based on a single GAS strain. There is no clarification of the representativeness of the strain. Two further strains were used for validation but these were incompletely described and are unlikely to be fully representative.
- e) For the epitope landscaping, it was not clear why the three proteins analysed were selected.
- f) Characterisation of the epitopes identified, particularly in terms of their 3D positioning on the protein were only done for a limited number. A more extensive analysis may have identified common themes.

3) DISCUSSION

The extent to which all the antibodies identified are likely to be cross-reactive to other streptococcal antigens was not discussed. Beyond the novelty of the methodology, it was not clear what novel biology was discovered. It is stated that the antigenome is on average 30 antigens/individual but this is based on just 10 participants. The limitations of these analyses are not clearly stated.

Reviewer #2

(Remarks to the Author)

This paper describes a new method for characterising circulating pathogen-specific antibodies centred in mass-

spectrometry, and the application of this to Streptococcal species.

The anti-genome aspects are very comprehensive and this study will no doubt provide a valuable resource for the field in terms of major immunodominant antigens identified by this approach. I found this part of the study and the epitope mapping aspect really impressive.

In the abstract and introduction the authors highlight how systems serology, which involves in depth characterisation of Fc, tends to be focused on known antigens. It suggested that this study provides a more global view in Fc function. However, the Fc characterisation presented is also focused on a small selected number of antigens identified by the anti-genomic approach - M protein, C5AP and PRGA. This is not a criticism of the study, rather the way it has been framed - particularly in the significance section and introduction. A more accurate description might be global interrogation of the anti-genome and focused analysis of identified antigens with respect to Fc characteristics.

Other comments for consideration:

1. Lines 112-113 - examples of systems serology, the most relevant from the literature is probably Strep pnemo, which is not included. Suggest this is added - Davies, L. R. L. et al. Polysaccharide and conjugate vaccines to Streptococcus pneumoniae generate distinct humoral responses. *Sci Transl Med* 14, eabm4065 (2022).
2. Lines 156-159 and throughout. The authors use acronyms for the Strep proteins identified and it was very difficult to find what these stand for or the full names/descriptions. Suggest a table or descriptor list is included somewhere in the main manuscript. Also noting that Spy_0747 has been named SpnA, and is very well characterised in the context of clinical serology for GAS - Whitcombe, A. L. et al. Development and evaluation of a new triplex immunoassay that detects Group A Streptococcus antibodies for the diagnosis of rheumatic fever. *J. Clin. Microbiol.* 58, 264 (2020).
3. The common set of 11 antigens - many of these have been identified in other approaches. The authors do reference these in their intro (eg Bencie et al and Reglinski et al), but dont clearly circle back to these in the discussion. That many of the antigens have now been identified across multiple countries with different IVIG and patient samples likely cements them as truly immunodominant and is worth more discussion.
4. M protein reactivity - lines 303-305 there is mention that bacteremia patients react with the conserved region more frequently. This maybe because they had non-M1 infections. It wasn't mentioned in the methods whether the infecting strain type from the bacteremia patients was known? If this is known it should be included as would help with interpretation.
5. The assays to quantify engagement of CD32 and CD16 produced somewhat unexpected results in that C5AP was more active than M1. However, the study then goes on to characterise M protein in detail, presumably because M protein is well documented as a major antigen with respect to phagocytosis. I have some concerns that these assays rely on the GAS antigen being coated onto an ELISA plate, this will immobilise the antigen and potentially restrict antibody binding and flexibility that will then allow for receptor dimerization. What about assays used in other systems serology analysis that have the antigen coupled to a bead that may circumvent these steric limitations?
6. The phagocytosis assays make use of THP-1 cells. These cells do not have complement receptors and as such they are somewhat limited in their replication of opsonophagocytosis in vivo. Might assays involving neutrophil like cells with complement receptors (like HI-60 cells) be considered for this work flow in the future? This will give a more full-some picture if Fc function, especially for bacterial antibodies where OPK is likely a critical function for protection.
7. The elevated anti-M IgG3 in septic patients has similarities with that recently observed in rheumatic fever in Lorenz et al., *iScience* 2024. These authors also suggest the flexibility of IgG3 may contribute to the enhanced anti-M binding. It is perplexing though that rheumatic fever is a post-infectious sequelae of GAS, and these children are not necessarily protected from future GAS infections. How might this fit with the suggestion in the discussion (lines 444-448) that anti-M IgG3 could be linked with more protective responses in adults vs children?

Reviewer #3

(Remarks to the Author)

Toledo et al. described an interesting work that a mass spectrometry-based antigenomics-serology workflow was established for investigation of circulating IgG against streptococcal pathogens. GAS-specific IgG circulating in human plasma was applied to isolate antigens from pools of bacterial proteins by using AP-MS, showing that GAS-specific IgG circulating in human plasma targeted a small subset of bacterial antigens. By further analyzing the plasma samples from healthy donors, they found that the individual antigenomes converged around a small subset of genomically conserved antigens, and that there were significant differences in pathogen-specific antibody responses during sepsis. They further profiled the antigenic sites frequently targeted by circulating GAS-antibodies and found that IgG subclass affected the capacity of anti-M1 antibodies to trigger immune signaling. Overall, the manuscript is well-written and most of the conclusions can be supported by the results. This reviewer has several concerns as shown below.

Major concerns:

1. The sections of Results and Introduction need to be re-constructed to show the innovation and significance of the method

development.

The title of this manuscript highlights the combined systems antigenomics-serology workflow, which should be the major contribution of this work. In Line 127-134 of Page 4, the authors described the two-step approach shown in Figure 1A for characterization of the structural and functional properties of circulating IgG antibodies against GAS in human bodies. Actually, it is an integrated workflow rather than a newly developed approach. From the description on this approach, it is difficult to understand why and how for method development. Therefore, this reviewer suggests to address: i) why these experiments are combined and organized for characterize the circulating IgG antibodies? ii) What is the major purpose for every experimental design? iii) What is the relationship or connection between the two steps? iv) What data can be used to validate the feasibility of the approach and what data for application scenario? v) In addition, if the major innovation of this work is based on the establishment of this integrated workflow shown in Figure 1A, what are the current technical challenges and application requirements, and what advantages and improvements in comparison to previous reports? Collectively, it is better for the authors to clearly address the current analytical challenges in characterization of circulating antibodies, the technical innovation in the current work, new findings in the current results, the therapeutic or clinical significances resulting from the immunological map data.

Minor concerns:

1. In the line 140 of page 4, how the proteins from secreted, cell wall and membrane were annotated?
2. In Figure 1G, H and Figure 3C, D, better to show the cut-off lines in the volcano plots for the fold changes and p-values.
3. In the line 284 of page 8, better to add more details on the HDX-MS experiment, since this step is important in the workflow. Why 75% of maximum theoretical uptake is used as a criterion?
4. In the line 705-725 of page 20, the data analysis on glycoproteomics need to be added. The mass error tolerance and missing cleavage for database search are missing.

Version 1:

Reviewer comments:

Reviewer #3

(Remarks to the Author)

The authors have answered most of the reviewer's questions. However, there are still some concerns that the authors need to address for improving the manuscript.

- 1.Regarding the response to the reviewer's major concern, the analytical needs on linking antigen specificity with functional antibody responses, are still weak for this integrated workflow. The author better to show some comparative data with previous reports to demonstrate the advantage of the current workflow. In addition, what is the therapeutic or clinical significances resulting from the immunological map data? What is the potential application scenario for this workflow?
- 2.Regarding the response to the reviewer's minor concern #1, better to validate the correctness of the protein annotations by using public database or published data.
- 3.Regarding the response to the reviewer's minor concern #3, please cite a reference that applies 75% of maximum theoretical uptake.

POINT-BY-POINT RESPONSE

Reviewer #1 (Remarks to the Author):

The authors have used what they term an integrative mass spectrometry-based approach which combines systems antigenomics and systems serology to characterize human antibodies to Group A Streptococcus (GAS). Unsurprisingly, both healthy and GAS-infected individuals have circulating IgG against genomically conserved streptococcal proteins. Again, as might be expected, there was inter-individual variation in titers, subclass distributions, and Fc signalling capacity. There was cross-reactivity with Streptococcus dysgalactiae. The novelty of this manuscript is in the integration of the technology.

Further comments

1) ABSTRACT:

a) There is insufficient detail for the reader to determine the precise nature of the technology or the results.

- We have re-written the abstract to better emphasize the methods and their expected outputs:

“This study showcases an integrative mass spectrometry-based strategy combining systems antigenomics and systems serology to characterize human antibodies in clinical samples. This strategy involves using antibodies circulating in plasma to affinity-enrich antigenic proteins in biochemically fractionated pools of bacterial proteins, followed by their identification and quantification using mass spectrometry. A selected subset of the identified antigens is then expressed recombinantly to isolate antigen-specific IgG, followed by characterization of the structural and functional properties of these antibodies.

We focused on Group A streptococcus (GAS), a major human pathogen lacking an approved vaccine. The data shows that both healthy and GAS-infected individuals have circulating IgG against conserved streptococcal proteins, including toxins and virulence factors. The antigenic breadth of these antibodies remains relatively constant across healthy individuals but changes considerably in GAS bacteremia. Moreover, antigen-specific IgG analysis revealed individual variation in titers, subclass distributions, and Fc-signaling capacity, despite similar epitope and Fc-glycosylation patterns. Finally, we show that GAS antibodies may cross-react with Streptococcus dysgalactiae (SD), a bacterial pathogen that occupies similar niches and causes comparable infections. Collectively, our results highlight the complexity of GAS-specific antibody responses, while showcasing the versatility of our methodology to characterize immune responses to bacterial pathogens.”

b) It is stated in the significance statement that “The method is cost effective, flexible, and adaptable” but this is not built on evidence provided in the manuscript.

- The reviewer is right since an objective cost analysis in comparison to other methods is not included in the paper. We have removed the significance statement altogether to avoid confusion and also because this section is not part of the standard formatting, as per the editorial guidelines of Nat Comm.

2) INTRODUCTION:

a) This provides a synopsis of IgG biology at a level not required for a manuscript of this nature

Since our target audience includes a broad spectrum of researchers working at the interface of host-pathogen interactions (e.g. microbiologists, structural biologists etc.), not just immunologists, we felt it necessary to introduce the key antibody features that we address in our analysis. These features include Fab binding, Fc structural diversity, signaling capacity, and the differences in complexity between monoclonal and polyclonal responses. Our methodology focuses on unraveling where these antibodies bind at both the antigen and epitope levels, their Fc structure, and the connection to FcR signaling, using both monoclonal antibodies and polyclonal sera.

b) Systems antigenomics and serology are reviewed at a very high level and not detail is provided.

- We have included some further details into previous approaches:

Antigenomics part:

“Examples include screening genome sequences of Group B Streptococcus, cloning surface-exposed antigens, and conducting immunization challenges in animal models; building protein arrays paired with flow cytometry binding assays to study antibodies against predicted surface GAS proteins; and using reverse vaccinology and human infection challenge models to explore the antigenic breadth of antibodies against the malarial parasite *Plasmodium falciparum*”.

Serology part:

“These studies have revealed that humoral responses elicited in four HIV vaccine trials result in unique humoral “Fc fingerprints”, that individuals with latent tuberculosis infection and active tuberculosis disease exhibit distinct MTB-specific humoral responses, and that specific Fcγ-receptor signaling plays a role in controlling SARS-CoV-2 infections, to only mention a few examples.”

2) METHODS & RESULTS:

a) An overall workflow and data integration image would have been useful.

- We included such an overview in Fig.1a. We have also added a sentence to the result section stating that:

An overview of the main components of the analytical strategy is shown in Fig. 1a.

b) The analysis of employed commercial sera from an undisclosed number of healthy donors and commercial IVIG, and from 4 patients with GAS bacteraemia and 5 patients with *S. dysgalactiae* bacteraemia (details not provided). Ten donors were used to determine inter-individual variation but these were incompletely described.

- The clinical details of the two patient cohorts have been published before and we added references to those papers in the material and method section, together with a clarifying statement on the healthy donors, for whom the main criteria of selection was the lack of obvious signs of infection and otherwise overall good health.

“Acute blood samples were collected within five days after hospital admission, and convalescent blood samples were collected after 4-6 weeks. Information on the included patients is given by *de Neergaard et.al* (26) and *Bläckberg et.al* (54). Sampling of healthy donor blood was approved by the regional ethics review board in Lund, Sweden (approval 2015/801). Citrated blood samples were collected from healthy donors without any sign of ongoing infection and otherwise in good health.”

c) It appears that both sera and citrated plasma were used but no data is providing as to the impact of the collection methods.

- We thank the reviewer for noting this typo. No sera was used in this study but only plasma. We have corrected the material and method section to acknowledge that.

d) Most of the analysis was based on a single GAS strain. There is no clarification of the representativeness of the strain. Two further strains were used for validation but these were incompletely described and are unlikely to be fully representative.

- The reviewer is correct in noting that no single genomic sequence can fully represent the genomic diversity of GAS, a limitation that applies to virtually any microbe. We acknowledge the limitation in the number of strains used in our study and agree that including additional strains could potentially broaden the antigen repertoire. However, the SF370 strain serves as a reference strain, as it was the first GAS strain to be fully sequenced and has been extensively studied in the context of GAS pathogenesis and vaccine development. We have added a comment to the results section to make our choice of strain clearer.

“This was the first fully sequenced strain of *S. pyogenes* and has served as a reference for studies exploring both GAS antigens as well as mechanism of GAS disease(24).”

e) For the epitope landscaping, it was not clear why the three proteins analysed were selected.

- We have now added more details to our rationale for antigen choice:

“To determine the epitope landscape of GAS-specific IgG, we implemented an epitope extraction (EpXT) workflow (Fig. 4a and Materials and Methods). We selected three antigens for this analysis: M1 and C5AP, identified in Cluster A, and PRG4, identified in Cluster B (Fig. 2a). M1 and C5AP were chosen because they are well-known virulence factors and vaccine candidates. In contrast, the function of PRG4 remains unknown, providing an opportunity to explore a less characterized protein. Structurally, M1 is a fibrillar protein, C5AP is a globular protein, and PRG4 is predicted to contain both fibrillar and globular regions, offering a diverse range of antigenic structures for analysis.”

f) Characterisation of the epitopes identified, particularly in terms of their 3D positioning on the protein were only done for a limited number. A more extensive analysis may have identified common themes.

- True, we are currently working on expressing more antigens and conducting further detailed analysis. We hope to complete these more comprehensive studies in the near future.

3) DISCUSSION

The extent to which all the antibodies identified are likely to be cross-reactive to other streptococcal antigens was not discussed.

- We modified the discussion to include potential cross-reactivity not only with SD but also with other streptococcal species, and the implications of these findings:

“Interestingly, the data also suggested potential immunological overlap between the GAS and SD antigenomes, based on the presence of homologous antigenic sequences across species. This overlap is most likely due to similarities in genetic composition resulting from frequent horizontal gene transfer (HGT), which increases the possibility of immunological cross-reactions(30, 31). There is some evidence that antibodies elicited against one bacterial pathogen can provide protection against another pathogen due to cross-reactivity(42–44). Whether similar mechanisms apply to GAS and SD, or even other streptococcal species, remains to be investigated. This is particularly important for vaccine design efforts, as targeting shared antigens might result in significant antimicrobial and epidemiological effects on both pathogen populations.”

Beyond the novelty of the methodology, it was not clear what novel biology was discovered.

-The study provides several novel biological insights. First, it identifies conserved antigenic signatures of GAS-specific IgG across healthy individuals and highlights shifts in antigen profiles during GAS bacteremia. Second, it reveals pathogen-specific antibody responses, despite genomic similarities between GAS and *Streptococcus dysgalactiae*, shedding light on cross-reactivity and its potential implications for vaccine design. Third, it demonstrates that IgG subclass distribution, particularly enrichment of IgG3, significantly impacts Fc γ -receptor signaling and immune activation, with implications for understanding protective immunity.

It is stated that the antigenome is on average 30 antigens/individual but this is based on just 10 participants. The limitations of these analyses are not clearly stated.

- We have now added a final paragraph that clearly states the limitations of this study:

“Finally, while this study provides significant insights into the antigenic and functional properties of GAS-specific IgG, several limitations should be acknowledged. The use of a limited number of GAS strains may not fully capture the genomic and antigenic diversity of the pathogen, and the small clinical cohort may restrict the generalizability of some findings. Additionally, the focus on IgG responses excludes other antibody isotypes, such as mucosal IgA, which are highly relevant to the immune response against GAS. Nonetheless, the robustness and flexibility of our integrated methodology provide a powerful framework to dissect antibody responses against complex pathogens. Future studies expanding strain diversity and clinical cohorts will build on these findings to further advance our understanding of streptococcal immunity and accelerate vaccine development.”

Reviewer #2 (Remarks to the Author):

This paper describes a new method for characterising circulating pathogen-specific antibodies centred in mass-spectrometry, and the application of this to Streptococcal species.

The anti-genome aspects are very comprehensive and this study will no doubt provide a valuable resource for the field in terms of major immunodominant antigens identified by this approach. I found this part of the study and the epitope mapping aspect really impressive.

- We thank the reviewer for this comment. We also believe this technology will be useful to obtain deeper molecular understanding of immune responses to bacterial pathogens.

In the abstract and introduction the authors highlight how systems serology, which involves in depth characterisation of Fc, tends to be focused on known antigens. It suggested that this study provides a more global view in Fc function. However, the Fc characterisation presented is also focused on a small selected number of antigens identified by the anti-genomic approach - M protein, C5AP and PRGA. This is not a criticism of the study, rather the way it has been framed - **particularly in the significance section and introduction**. A more accurate description might be global interrogation of the anti-genome and focused analysis of identified antigens with respect to Fc characteristics.

We have now rephrased the introduction and result section to clarify the scope of our technique in accordance with the comments from the reviewer:

“The approach is based on streamlining antigen/antibody affinity purification workflows using an automated liquid-handling platform, coupled to a suite of high-resolution MS-based quantitative, structural, and glyco-proteomics readouts. The first step (systems antigenomics) focuses on antigen discovery, leveraging antigen/antibody affinity purification to isolate and identify GAS-specific antigens from bacterial protein fractions using high-resolution mass spectrometry. In the second step (systems serology), we focus on a subset of these identified antigens for detailed characterization of the corresponding antibody responses, integrating systems serology techniques to analyse functional and structural properties of the antigen-specific antibodies. The antigenome data then informs the selection of key antigens, enabling a focused analysis of their associated antibody responses. An overview of the main components of this analytical strategy is shown in Fig. 1a.”

Other comments for consideration:

1. Lines 112-113 - examples of systems serology, the most relevant from the literature is probably Strep pnemo, which is not included. Suggest this is added - Davies, L. R. L. et al. Polysaccharide and conjugate vaccines to Streptococcus pneumoniae generate distinct humoral responses. *Sci Transl Med* 14, eabm4065 (2022).

- We thank the reviewer for the suggestion. This reference has now been added.

2. Lines 156-159 and throughout. The authors use acronyms for the Strep proteins identified and it was very difficult to find what these stand for or the full names/descriptions. Suggest a table or descriptor list is included somewhere in the main manuscript. Also noting that Spy_0747 has been named SpnA, and is very well characterised in the context of clinical serology for GAS - Whitcombe, A. L. et al. Development and evaluation of a new triplex immunoassay that detects Group A Streptococcus antibodies for the diagnosis of rheumatic fever. *J. Clin. Microbiol.* 58, 264 (2020).

- The specific acronyms used throughout the manuscript are explained in Supplementary file S2, and we added now an explanatory comment in the text:

“(see Supplementary file S2, which includes descriptions of the acronyms for the selected GAS proteins used throughout the manuscript).”

3. The common set of 11 antigens - many of these have been identified in other approaches. The authors do reference these in their intro (eg Bensie et al and Reglinski et al), but dont clearly circle back to these in the discussion. That many of the antigens have now been identified across multiple countries with different IVIG and patient samples likely cements them as truly immunodominant and is worth more discussion.

- We agree with the reviewer and have now included a comment on this issue in the discussion:

“Our data confirm that all adult donors and patients included in this study have circulating IgG against the GAS antigenome, a subset of streptococcal antigens that are genomically conserved across GAS isolates. This antigenome covers a wide range of bacterial proteins, including many virulence factors. Notably, we identified a common set of 11 antigens that were consistently targeted by circulating IgG across healthy individuals and patients with GAS bacteremia. Many of these antigens, such as M1 and C5AP, have also been identified in previous studies using different experimental approaches. These repeated findings, now validated across studies performed in multiple countries, using distinct IVIG and patient sample cohorts, strongly reinforce the notion that these antigens are truly immunodominant, highlighting their potential relevance as vaccine candidates or as biomarkers for GAS exposure.”

4. M protein reactivity - lines 303-305 there is mention that bacteremia patients react with the conserved region more frequently. This maybe because they had non-M1 infections. It wasn't mentioned in the methods whether the infecting strain type from the bacteremia patients was known? If this is known it should be included as would help with interpretation.

- We referred to a previous paper where these patients were characterized in detail but we have now explicitly added the serotype information to the result section as well:

“We have previously reported the clinical and serological status of these patients, two of the patients were infected with emm1 isolates, and two with emm118 or emm85, but no major differences were found between their acute and convalescent plasma(27).”

5. The assays to quantify engagement of CD32 and CD16 produced somewhat unexpected results in that C5AP was more active than M1. However, the study then goes on to characterise M protein in detail, presumably because M protein is well documented as a major antigen with respect to phagocytosis. I have some concerns that these assays rely on the GAS antigen being coated onto an ELISA plate, this will immobilise the antigen and potentially restrict antibody binding and flexibility that will then allow for receptor dimerization. What about assays used in other systems serology analysis that have the antigen coupled to a bead that may circumvent these steric limitations?

- It is certainly possible that antigen presentation may differ between systems that immobilize proteins by coating onto a plate versus conjugating them to beads. However, it is important to note that the signaling assay results were consistent with the phagocytosis assays, at least for M1. Nonetheless, we have added a comment to the discussion to acknowledge this potential caveat.

“In this context, it is important to note that the signaling assay used in this study relies on coating antigens onto a plate, which may restrict antibody binding and flexibility, potentially affecting receptor dimerization. Exploring alternative systems, such as immobilizing antigens on beads or similar platforms to circumvent potential steric hindrances, could provide additional insights.”

6. The phagocytosis assays make use of THP-1 cells. These cells do not have complement receptors and as such they are somewhat limited in their replication of opsonophagocytosis in vivo. Might assays involving neutrophil like cells with complement receptors (like HI-60 cells) be considered for this work flow in the future? This will give a more full-some picture if Fc function, especially for bacterial antibodies where OPK is likely a critical function for protection.

- That is totally correct. We have now added a comment on this important issue:

“Similarly, complementing the signaling assays with phagocytosis assays using other cell lines that, unlike the THP1 cell line, express complement receptors could provide a more comprehensive understanding of Fc function, especially for bacterial antibodies where opsonophagocytic killing (OPK) is likely critical for protection.”

7. The elevated anti-M IgG3 in septic patients has similarities with that recently observed in rheumatic fever in Lorenz et al., iScience 2024. These authors also suggest the flexibility of IgG3 may contribute to the enhanced anti-M binding. It is perplexing though that rheumatic fever is a post-infectious sequelae of GAS, and these children are not necessarily protected from future GAS infections. How might this fit with the suggestion in the discussion (lines 444-448) that anti-M IgG3 could be linked with more protective responses in adults vs children?

- We thank the reviewer for bringing this recent paper to our attention. These apparently conflicting observations clearly highlight the need for better correlates of GAS immunity, including markers of susceptibility to autoimmune sequelae. We have now included this additional reference and added a comment to the text:

“At the same time, elevated anti-M IgG3 levels have recently been observed in patients suffering from acute rheumatic fever(55). These apparently conflicting observations highlight the need for better immune correlates of GAS infection, protection and susceptibility to autoimmune sequelae.”

Reviewer #3 (Remarks to the Author):

Toledo et al. described an interesting work that a mass spectrometry-based antigenomics-serology workflow was established for investigation of circulating IgG against streptococcal pathogens. GAS-specific IgG circulating in human plasma was applied to isolate antigens from pools of bacterial proteins by using AP-MS, showing that GAS-specific IgG circulating in human plasma targeted a small subset of bacterial antigens. By further analyzing the plasma samples from healthy donors, they found that the individual antigenomes converged around a small subset of genomically conserved antigens, and that there were significant differences in pathogen-specific antibody responses during sepsis. They further profiled the antigenic sites frequently targeted by circulating GAS-antibodies and found that IgG subclass affected the capacity of anti-M1 antibodies to trigger immune signaling. Overall, the manuscript is well-written and most of the conclusions can be supported by the results. This reviewer has several concerns as shown below.

Major concerns:

1. The sections of Results and Introduction need to be re-constructed to show the innovation and significance of the method development.

The title of this manuscript highlights the combined systems antigenomics-serology workflow, which should be the major contribution of this work. In Line 127-134 of Page 4, the authors described the two-step approach shown in Figure 1A for characterization of the structural and functional properties of circulating IgG antibodies against GAS in human bodies. Actually, it is an integrated workflow rather than a newly developed approach. From the description on this approach, it is difficult to understand why and how for method development. Therefore, this reviewer suggests to address: i) why these experiments are combined and organized for characterize the circulating IgG antibodies? ii) What is the major purpose for every experimental design? iii) What is the relationship or connection between the two steps? iv) What data can be used to validate the feasibility of the approach and what data for application

scenario? v) In addition, if the major innovation of this work is based on the establishment of this integrated workflow shown in Figure 1A, what are the current technical challenges and application requirements, and what advantages and improvements in comparison to previous reports? Collectively, it is better for the authors to clearly address the current analytical challenges in characterization of circulating antibodies, the technical innovation in the current work, new findings in the current results, the therapeutic or clinical significances resulting from the immunological map data.

- We thank the reviewer for pointing out a need for a more pronounced emphasis on the technical needs of the field and the advances introduced by our study. We have modified both introduction, and result sections to clarify that:

“Mass spectrometry (MS) is a highly sensitive and versatile analytical method for protein identification, quantification, and the characterization of post-translational modifications and protein-protein interactions. Despite advances in systems antigenomics for unbiased antigen discovery and systems serology for functional antibody analysis, these methods are often applied independently, leaving critical gaps in linking antigen specificity with functional antibody responses. The flexibility of modern MS technologies now allows for the integration of these complementary approaches. Here, we present an automated and quantitative workflow that combines systems antigenomics and systems serology to provide an extensive, multilayered characterization of antigen-specific antibodies. Compared to previous methods, this workflow identifies relevant antigens in an unbiased manner while simultaneously analyzing antibody titers, epitope landscapes, subclass distributions, and Fc-mediated functions. We applied this approach to study antibody responses against Group A Streptococcus (GAS), a major bacterial pathogen and significant source of global morbidity and mortality worldwide(22).”

“The approach is based on streamlining antigen/antibody affinity purification workflows using an automated liquid-handling platform, coupled to a suite of high-resolution MS-based quantitative, structural, and glyco-proteomics readouts. The first step (systems antigenomics) focuses on antigen discovery, leveraging antigen/antibody affinity purification to isolate and identify GAS-specific antigens from bacterial protein fractions using high-resolution mass spectrometry. In the second step (systems serology), we focus on a subset of these identified antigens for detailed characterization of the corresponding antibody responses, integrating systems serology techniques to analyse functional and structural properties of the antigen-specific antibodies. The antigenome data then informs the selection of key antigens, enabling a focused analysis of their associated antibody responses. An overview of the main components of this analytical strategy is shown in Fig. 1a.”

Minor concerns:

1. In the line 140 of page 4, how the proteins from secreted, cell wall and membrane were annotated?

- We understand the reviewer is asking how we annotated the subcellular localizations. In this case, they were annotated based on the experimental information itself, in other words, based on in which biochemical fraction/s the antigens were experimentally identified.

2. In Figure 1G, H and Figure 3C, D, better to show the cut-off lines in the volcano plots for the fold changes and p-values.

- We have corrected the figure.

3. In the line 284 of page 8, better to add more details on the HDX-MS experiment, since this step is important in the workflow.

- We have now added more information regarding the methodological principles and rationale of the HDX experiments:

“To validate these immunodominant regions, we performed hydrogen-deuterium exchange mass spectrometry (HDX-MS), a technique that measures the exchange of hydrogen atoms with deuterium in a protein's backbone amide groups. The rate of this exchange is influenced by the structural flexibility and solvent accessibility of the protein, with regions buried within the protein structure or bound by antibodies exhibiting slower exchange rates. HDX-MS identified two peptide stretches (97-138 aa and 415-466 aa) that displayed a significant reduction in deuterium uptake upon incubation with IVIG (Fig. 4c & Supplementary file S7), indicating that these regions were engaged by antibody binding. These binding sites partially overlapped with those identified by the EpXT workflow, demonstrating good agreement between the methods and further validating the Cat-domain as an immunodominant region (Fig. 4d).

Why 75% of maximum theoretical uptake is used as a criterion?

- In our system, as in others, achieving 100% deuteration is not feasible due to two primary factors: a dilution with D₂O of approximately 1:10 and back-exchange occurring within the system. To account for this, we set the maximum deuteration level to 75% and ensure that no peptide exceeds 100% theoretical uptake. While this approach may result in slightly over- or underestimation of actual uptake values, it does not affect the validity of our results, as our analysis is based on relative comparisons of uptake between bound and unbound states. This method allows us to obtain accurate and reliable data without the need for a fully deuterated control, while maintaining consistency in the relative differences observed.

4. In the line 705-725 of page 20, the data analysis on glycoproteomics need to be added. The mass error tolerance and missing cleavage for database search are missing.

- We thank the reviewer and have added now the requested information to the method section.

Point by point response:

Reviewer #3 (Remarks to the Author):

The authors have answered most of the reviewer's questions. However, there are still some concerns that the authors need to address for improving the manuscript.

1.Regarding the response to the reviewer's major concern, the analytical needs on linking antigen specificity with functional antibody responses, are still weak for this integrated workflow. The author better to show some comparative data with previous reports to demonstrate the advantage of the current workflow.

- We thank the reviewer for raising this point. We would like to point out that linking antigen specificity to the immunological properties of antibodies has not been previously achieved for GAS. It is consequently challenging to compare our data with previous reports. Addressing this gap is in fact a central rationale for our study. Our results demonstrate the strength of our integrated workflow in addressing the critical gap in linking antigen specificity with functional antibody properties, as highlighted by the reviewer's valuable feedback. We have now added a paragraph to the discussion regarding this issue:

"Finally, a key contribution of this study lies in its ability to link antigen specificity with functional antibody properties, which is poorly understood in the context of GAS immunity. This is particularly evident in the analysis of antibody responses against the M1 protein, where we identified that M1-specific antibodies target immunodominant regions, including the hypervariable and variable regions, while exhibiting subclass-specific functional properties. For example, anti-M1 antibodies from septic patients displayed an increased proportion of IgG3, which was associated with enhanced Fcγ-receptor engagement and opsonophagocytic activity. These findings highlight the strength of our integrated workflow in coupling antigenic targets to immune functions, to start addressing this longstanding gap in GAS immunology. "

In addition, what is the therapeutic or clinical significances resulting from the immunological map data? What is the potential application scenario for this workflow?

- The immunological mapping provided by our workflow is useful to identify conserved and immunodominant antigens that are potential candidates for vaccine development or as biomarkers for exposure to GAS. It also reveals changes in the antigenome during GAS bacteremia, suggesting its utility in understanding immune responses during infections. The workflow is particularly suited for vaccine development, where identifying conserved immunodominant antigens is critical. In fact, we are currently using the top ranking antigens as targets for vaccine development programs ongoing in the lab. We further plan to use this methodology to extend the antigen landscape for GAS and for other pathogens to generate a data-driven resource for antigens. Lastly, the method could also be applied in diagnostics to differentiate healthy states from GAS infection or to explore immune responses to other bacterial pathogens by adapting the method to different bacterial strains or clinical cohorts. Additionally, it offers a framework for studying cross-reactivity with related pathogens, such as *Streptococcus dysgalactiae*.

2.Regarding the response to the reviewer's minor concern #1, better to validate the correctness of the protein annotations by using public database or published data.

- In response to the reviewer's concern, we have cross-checked the subcellular localizations against the information available on UniProt and updated Supplementary File S1 with an additional column for the SF370 strain, one of the few strains that has previously been interrogated with proteomics.

In general, both the GAS proteome and its subcellular localization are poorly characterized. This issue is further complicated by the presence of many proteins located across multiple subcellular locations and possessing moonlighting functions. In this context, our data provide actual experimental evidence of the localization of these proteins under the conditions of this study.

3.Regarding the response to the reviewer's minor concern #3, please cite a reference that applies 75% of maximum theoretical uptake.

- We have added a comment to the method section referring to the rationale for our choice of maximum theoretical uptake and added three references, two referring to the well-known problem with back exchange, and another referring to a paper where this cut-off was also applied

"In our system, achieving 100% deuteration is not feasible due to two primary factors: a dilution with D₂O of approximately 1:10 and back-exchange occurring within the system(57, 58). To account for this, we set the maximum deuteration level to 75% and ensure that no peptide exceeds 100% theoretical uptake(59). While this approach may result in slightly over- or underestimation of actual uptake values the analysis is still based on relative comparisons of uptake between bound and unbound states."